# Revisiting Dynamic Graphs from the Perspective of Time Series

## Abstract

Numerous studies have investigated temporal modeling in dynamic graphs. Existing approaches predominantly fall into two categories: discrete-time dynamic graph (DTDG) methods and continuous-time dynamic graph (CTDG) methods. While both paradigms have shown effectiveness in capturing temporal dependencies, they suffer from several inherent limitations. Specifically, DTDG approaches often lose fine-grained temporal information due to snapshot-based discretizations, whereas CTDG methods preserve precise timestamps but may struggle to capture long-range temporal dependencies because of computational constraints. Moreover, interactions in real-world dynamic graphs frequently exhibit predictable and recurring temporal patterns, which are not fully exploited by existing methods. To better leverage such regularities, we propose to transform node interactions into binary time-series representations, enabling explicit modeling of temporal patterns. Building on this formulation, we introduce a novel model, termed **T**ime **S**eries-based **Dy**namic **G**raph (TSDyG), which approaches dynamic graph learning from a time-series perspective. Compared to existing DTDG and CTDG methods, TSDyG offers several advantages: it preserves fine-grained temporal information, captures long-range dependencies, and effectively capture recurring interaction patterns. We conduct extensive experiments on multiple benchmark datasets, and the results demonstrate that TSDyG achieves competitive performance on downstream tasks such as temporal link prediction.

## 1 Introduction

Dynamic graphs model evolving systems in which interactions between entities change over time. Many real-world scenarios, such as social networks, user-item interactions, and financial transactions, can be naturally represented as dynamic graphs. In recent years, a growing number of research (Zhang et al., 2024; 2023; Ji et al., 2024; Cong et al., 2023) on dynamic graph learning has emerged, demonstrating its effectiveness in capturing temporal relationships among entities and achieving promising results in forecasting tasks.

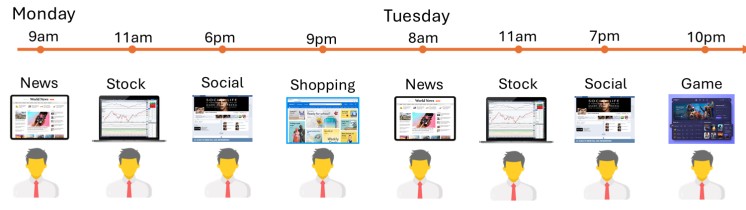

Figure 1: Illustration of regular temporal interaction patterns in a user's web-browsing history. For instance, the user often watch news and market website in the morning and social networks in the afternoon.

Current dynamic graph learning methods can generally be categorized into two types: discrete-time dynamic graph (DTDG) methods (Karmim et al., 2024; You et al., 2022; Yang et al., 2021; Sankar et al., 2020; Pareja et al., 2020) and continuous-time dynamic graph (CTDG) methods (Yu et al., 2023; Tian et al., 2023; Zhang et al., 2024; 2023; Ji et al., 2024; Zou et al., 2024; Poursafaei et al., 2022; Gravina et al., 2024).

In DTDG methods, the dynamic graph is represented as a sequence of snapshots that are in the form of static graphs to capture the interactions of entities during the specific time interval. These models typically employ Graph Neural Networks (GNNs) (Kipf & Welling, 2017; Hamilton et al., 2017; Xu et al., 2019) in conjunction with Recurrent Neural Networks (RNNs) (Hochreiter & Schmidhuber, 1997; Cho et al., 2014) to capture both the structural and temporal dependencies in the evolving graph. However, DTDG methods exhibit several limitations (Fennell et al., 2016; Cho et al., 2014). First, due to the partitioning of interactions into discrete snapshots, fine-grained temporal information is lost, which can negatively impact performance in time-sensitive prediction tasks. Second, selecting an appropriate snapshot interval is non-trivial: if the interval is too small, it can lead to redundant and computationally expensive graph sequences; if too large, important temporal details may be overlooked.

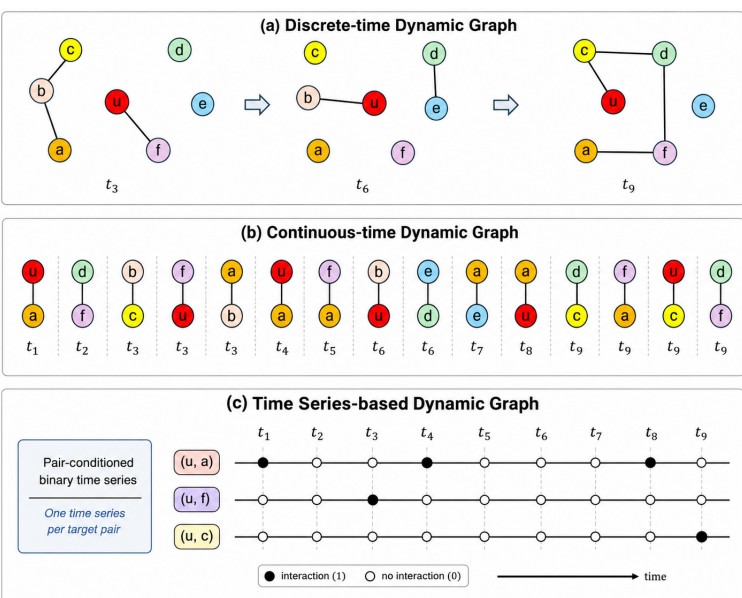

Figure 2: The illustration of discrete-time dynamic graph, continuous-time dynamic graph and our proposed time series-based dynamic graph.

Continuous-time dynamic graph (CTDG) methods, in contrast to DTDG approaches, represent dynamic graphs as sequences of chronologically ordered events (Yu et al., 2023; Wang et al., 2021a). Two main categories of CTDG methods have been developed: model-centric and memory-based approaches. Compared to DTDG methods, CTDG models can better preserve fine-grained temporal information. However, CTDG methods also face several limitations. Model-centric approaches (Yu et al., 2023; Zou et al., 2024; Wu et al., 2024) , such as those based on Transformers, often struggle to capture long-range temporal dependencies due to their high computational complexity over continuous event streams. On the other hand, memory-based methods (Ji et al., 2024; Su et al., 2024; Rossi et al., 2020), typically exhibit inferior performance because they process batches of events concurrently rather than sequentially, violating the natural chronological order of interactions, a challenge often referred to as temporal discontinuity (Su et al., 2024).

In real-world dynamic graphs, interactions between nodes are not merely stochastic; instead, they frequently exhibit temporal regularities and recurring behavioral patterns. As illustrated in Figure 1, a user's web-browsing history typically follows a routine-driven process: accessing financial news and stock websites during morning hours, and shifting toward social media and gaming platforms in the afternoon. Such behaviors reflect latent temporal patterns governed by diurnal cycles and contextual time cues. Capturing these patterns is essential for modeling the complex temporal dependencies that drive the evolution of dynamic graphs. A key observation is that these interaction patterns are intrinsically tied to their respective timestamps, implying that the likelihood of an interaction depends not only on the order of historical events but also on the specific time at which they occur. For example, at a future timestamp $t'$ (occurring in either the morning or afternoon), we want to know if a man watches news websites; in this context, the act of watching news would be distinguished from watching other websites, and the entire history of interactions can be viewed as a signal where the probability of a link is a continuous function of time rather than merely a sequence of discrete, isolated events. This perspective motivates us to shift the paradigm: rather than treating dynamic graphs as sequences of edges or snapshots of static graphs, we propose to perceive historical interactions as time-series data. By transforming these interactions into time-series representations, we can leverage advanced time-series modeling techniques to explicitly capture the trends and regular patterns inherent in dynamic graphs. Specifically, given a sequence of interactions $\mathcal{G} = \{(u_1, v_1, t_1), (u_2, v_2, t_2), \ldots), (u_T, v_T, t_T)\}$ with $0 \le t_1 \le t_2 \le \cdots \le t_T$, for the source node $u$ and a destination node $v$, their past interactions over time can be represented by the proposed function $f_{u,v}(t)$, which captures the interaction dynamics as a function

of time and can be defined as:

$$f_{u,v}(t) = \begin{cases} 1, & \text{if } (u,v,t) \text{ or } (v,u,t) \in \mathcal{G} \\ 0. & \text{otherwise.} \end{cases} \tag{1}$$

This binary time series $\{f_{u,v}(t)\}_{t=t_1}^{t_T}$ encapsulates the complete interaction history between $u$ and $v$. In our formulation, '1' denote the interactions for the source node $u$ and destination node $v$, while '0' denote the interaction between node $u$ and other nodes. The binary values are adopted to differentiate the interactions between the target node pairs and interactions between non-target node pairs. The difference of discrete-time dynamic graphs, continuous-time dynamic graphs and our proposed time series-based dynamic graphs are illustrated in Figure 2.

Previous work, such as FreeDyG (Tian et al., 2023), has highlighted the importance of recurring and periodic interaction patterns for dynamic graph modeling. FreeDyG (Tian et al., 2023) primarily captures evolving and shifting temporal patterns through frequency-domain enhancement. In contrast, our method does not rely on Fourier transformation or frequency-domain filtering. Instead, we introduce a new pair-conditioned time-series representation of dynamic graphs and develop a graph-aware model that learns temporal evolution directly in the time domain. Specifically, the proposed representation preserves the ordered interaction history associated with a target node pair while distinguishing interactions within the target pair from those involving alternative nodes. This pair-specific formulation offers a complementary perspective on dynamic graph modeling by explicitly characterizing how the interaction pattern of a given node pair evolves relative to its surrounding temporal context.

Enlightened by this perspective, we propose a novel dynamic graph learning method named **T**ime **S**eries-based **Dy**namic **G**raph (TSDyG) model which handles the dynamic graph from the perspective of time series. TSDyG comprises three key components: a time series formulation module, an embedding generation module, and a cross-attention module. In the time series formulation module, we convert node interactions into binary time series as defined in Eq. 1. Each time step indicates interactions for the target node pair (1) or interactions for other nodes (0). Next, the embedding generation module employs a projector to generate the interaction embeddings from the binary time series data, To incorporate temporal information, a time encoder is adopted to generate time-specific embeddings. To enhance the expressiveness of the representations and incorporate richer structural information, we concatenate edge embeddings and co-occurrence frequency embeddings with interaction and temporal embeddings to construct the input to the cross-attention module. The cross-attention module, drawing inspiration from prior work (Kim et al., 2024), introduces a learnable query token that interacts with the key-value pairs derived from the input embeddings. This design facilitates the modeling of long-range temporal dependencies from historical data while maintaining lower computational complexity compared to traditional self-attention mechanisms. During training, our model is optimized with the binary cross-entropy (BCE) loss. Compared to the previous DTDG and CTDG methods, TSDyG is distinguished by its ability to leverage both existing and non-existing interactions to model the recurring interaction patterns among nodes, while effectively capturing long-term dependencies in dynamic graphs. The contributions of our paper are summarized as follows:

- Unlike previous DTDG and CTDG methods that treat dynamic graphs as sequences of snapshots or discrete events, we introduce a novel formulation that represents dynamic graphs as time series. This formulation captures the regularities in the interactions of target node pairs, offering a unique perspective on node dynamics.

- Building on the formulated binary time series data, we propose the Time Series-based Dynamic Graph (TSDyG) model, which comprises three key components. In contrast to previous dynamic graph methods, TSDyG effectively captures recurring interaction patterns between nodes and models long-term temporal dependencies in dynamic graphs.

- Beyond the proposed TSDyG model itself, our time-series formulation broadens the methodological scope of dynamic graph research and establishes a meaningful bridge between dynamic graph learning and time-series modeling, thereby enabling techniques developed for time-series analysis to be explored in dynamic graph settings.

- We extensively evaluate our model on multiple benchmark datasets, and the results demonstrate that it achieves competitive performance on downstream tasks, such as link prediction, compared to the baselines.

## 2 Related Work

**Dynamic Graph Learning.** Existing methods can be roughly categorized into discrete-time and continuous-time approaches. Discrete-time methods (Wang et al., 2025c; Karmim et al., 2024; You et al., 2022; Yang et al., 2021; Sankar et al., 2020) regard dynamic graphs as a sequence of snapshots taken at regular time intervals, and typically extend the graph neural networks for static graphs to capture the temporal correlations. Recent work (Karmim et al., 2024) has explored graph transformers as a powerful alternative to GNN for modeling node dependencies. Another work (Zhang et al., 2025) proposed a temporal consistency-aware augmentation framework for dynamic graphs to address the inability of existing methods to preserve temporal consistency. However, discrete-time methods usually suffer significant limitations, such as the loss of temporal information. In contrast, continuous-time methods (Li et al., 2025; Zhang et al., 2024; Zou et al., 2024; Poursafaei et al., 2022) represent dynamic graphs as the chronologically ordered sequences of events. Among the continuous-time methods, memory-based methods (Liu et al., 2025; Ji et al., 2024; Su et al., 2024; Rossi et al., 2020) maintain a memory to update the node states based on interactions. However, during batch processing, the strict chronological order of the events may be violated. Model-centric methods (Yu et al., 2023; Zou et al., 2024; Wu et al., 2024) leverage sequential models such as LSTMs (Hochreiter & Schmidhuber, 1997), Transformers (Vaswani et al., 2017), MLP-Mixers (Tolstikhin et al., 2021)) and state space model (Gu et al., 2022) to capture long-range node dependencies while aiming to reduce the time complexity. Among them, FreeDyG (Tian et al., 2023) considers recurring and shifting interaction patterns in dynamic graphs, but it remains within the conventional continuous-time event-sequence framework and models such patterns through frequency-domain enhancement. In contrast, our method introduces a pair-conditioned time-series formulation that explicitly distinguishes interactions within a target node pair from those involving alternative nodes. Rather than relying on Fourier transformation or frequency filtering, it models the ordered pair-specific interaction sequence directly in the time domain, providing a complementary perspective on temporal regularity in dynamic graphs. Other methods have proposed techniques like temporal walk (Lu et al., 2024; Wang et al., 2021b; Jin et al., 2022) and graph ordinary differential equation (graph ODE) (Wang et al., 2025b; Gravina et al., 2024; Luo et al., 2023) for dynamic graph representation learning. Additionally, several studies (Yuan et al., 2024; Yang et al., 2024) have shown that existing dynamic graph methods often struggle to generalize under distribution shifts, prompting the development of new methods to address these challenges.

**Time Series Forecasting.** Time series forecasting is one of the fundamental tasks in time series analysis. Traditional statistical approaches, such as VAR (Watson, 1994) and ARIMA (Box et al., 1974) are often inadequate when dealing with non-linear temporal dynamics. In contrast, deep learning methods have demonstrated strong capabilities in capturing complex temporal patterns. Based on their architectural backbones, these methods can be broadly classified into four categories: CNN-based, RNN-based, Transformer-based, and MLP-based models. CNN-based methods (Liu et al., 2022) utilize convolution kernels to model local temporal variations. However, due to their limited receptive fields, they struggle to capture long-term dependencies. RNN-based methods (Salinas et al., 2020; Lai et al., 2018) model the temporal state Transition via recurrent structure. In comparison, transformer-based methods (Kitaev et al., 2020; Zhou et al., 2021; Kim et al., 2024; Liu et al., 2024; Nie et al., 2023; Zhang & Yan, 2023) achieve superior performance in forecasting tasks by introducing techniques like patching for efficient modeling of long-range dependencies. More recently, inspired by the MLP-based method (Zeng et al., 2023; Wang et al., 2024), recent work (Kim et al., 2024) further demonstrates that cross-attention is more effective than self-attention in time series forecasting. Beyond time-domain approaches, there is also a growing body of work (Zhou et al., 2022; Wang et al., 2025a; Eldele et al., 2024; Yi et al., 2023) focusing on frequency-domain modeling, which seeks to capture temporal patterns using spectral techniques. These frequency-aware methods (Zhou et al., 2022; Wang et al., 2025a; Eldele et al., 2024) have achieved competitive results and offer a complementary perspective to traditional time-domain forecasting models.

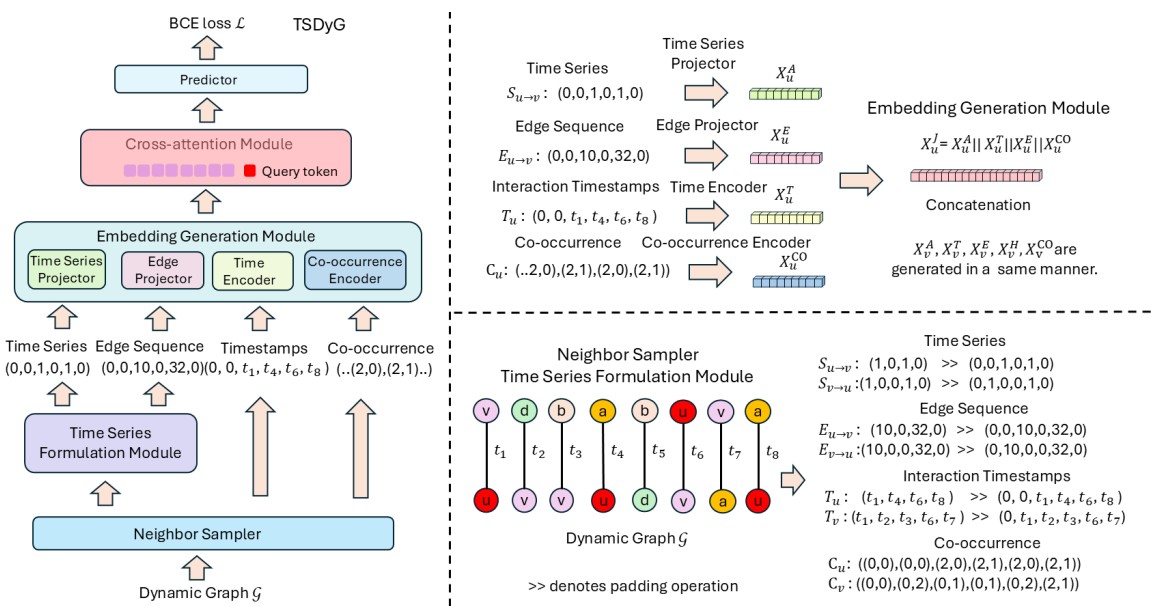

Figure 3: The overview of the proposed Time Series-based Dynamic Graph (TSDyG) model. TSDyG comprises three main components: (1) the time series formulation module, which generates binary time series from dynamic graphs; (2) the embedding generation module and (3) the cross-attention module, which models temporal evolution by extracting informative patterns from the time series.

## 3 Preliminary

**Discrete-time Dynamic Graph (DTDG).** The discrete-time dynamic graph is represented as a sequence of snapshots $\mathcal{G} = \{G_1, G_2, \dots\}$, where each snapshot $G_t = (\mathcal{V}_t, \mathcal{E}_t)$ is a static graph sampled at regular time intervals. $\mathcal{V}_t \subseteq \mathcal{V}$ denotes the set of active nodes at timestamp $t$, where $\mathcal{V}$ is the complete node set, and $\mathcal{E}_t \subseteq \mathcal{V} \times \mathcal{V}$ represents the set of observed edges at timestamp $t$.

**Continuous-time Dynamic Graph (CTDG).** The continuous-time dynamic graph usually consists of non-decreasing chronological events $\mathcal{G} = \{(u_1, v_1, t_1), (u_2, v_2, t_2), \dots, (u_T, v_T, t_T)\}$, where $0 \le t_1 \le t_2 \le \cdots \le t_T$. Each triplet $(u_i, v_i, t_i)$ signifies an interaction between source node $u_i \in \mathcal{V}$ and destination node $v_i \in \mathcal{V}$ at timestamp $t_i$.

**Time series-based Dynamic Graph (TSG).** We define a time series-based dynamic graph by converting node interactions into binary time series. For each pair node $(u, v) \in \mathcal{V} \times \mathcal{V}$, we define its interaction series as $\{f_{u,v}(t)\}_{t=t_1}^{t_T}$, where $f_{u,v}(t) \in \{0, 1\}$ indicates whether an interaction occurred between node $u$ and $v$ at timestamp $t$. The function $f_{u,v}(t)$ is formally defined in Eq. 1.

For attributed dynamic graphs, each interaction $(u, v, t)$ is associated with an edge feature $e_{u,v}^t \in \mathbb{R}^{d_E}$, where $d_E$ denotes the dimension of the edge feature. If the graph is non-attributed, the edge feature is simply set to zero vectors.

**Problem Formalization.** Given the formulated time series of the source node $u$ and destination node $v$ and historical interactions prior to timestamp $t$, i.e., $\{(u', v', t')|t' < t\}$, representation learning on the time series-based dynamic graphs aims to develop a model that convert the historical interactions into time series for target source node $u$ and destination node $v$ and learns time-aware representations that capturing the temporal patterns of their interactions. The effectiveness of the learned representation is evaluated through the link prediction, which predicts whether $u$ and $v$ are connected at future timestamp $t$.

## 4 Method

In this section, we introduce our proposed TSDyG. TSDyG is composed of three core components: a time series formulation module, an embedding generation module, and a cross-attention module. The overall architecture of TSDyG is illustrated in Figure 3.

**Time Series Formulation Module.** Given the historical interactions of source node $u$ and destination node $v$, the time series formulation module aims to construct the time series leading up to the current timestamp $t_c$. We sample the timestamps at which actual interactions involving either the source or destination node occur. Specifically, for source node $u$, we define the interaction timestamps as $T_u = \{t|(u, o, t)$ or $(o, u, t) \in \mathcal{G}, o \in \mathcal{V}, t < t_c\}$. For efficient batch processing, we retain the most recent $N$ timestamps from $T_u$ to ensure that the constructed time series both tractable and informative. Using these timestamps, we construct a binary time series sequence for source node $u$ with respect to neighboring node $v$, denoted as $S_{u \to v} = f_{u,v}(T_u) \subseteq \{0, 1\}^N$, where each entry indicates whether an interaction occurs between nodes $u$ and $v$ at the corresponding timestamp. For example, suppose source node $u$ had historical interactions from $t_1$ to $t_6$. and only interacted with destination node $v$ at $t_3$ and $t_5$. Then, the resulting binary time series would be $\{0, 0, 1, 0, 1, 0\}$. If the sequence length is shorter than $N$, zero-padding is applied to maintain a consistent length.

Unlike discrete-time dynamic graph methods, our approach does not partition the interaction stream into fixed-width snapshots or sample uniformly spaced time points. For each target pair, it constructs a pair-conditioned sequence from all actual historical events involving the source node and its neighbors. while preserving the timestamp and irregular interval associated with every event through explicit temporal encoding. Consequently, the method neither requires selecting a snapshot interval nor changes the temporal resolution of the original graph, and no interaction is omitted simply because it occurs between predefined sampling points. This event-driven formulation enables the model to capture recurring and periodic patterns from both the ordering and timing of interactions, while also retaining infrequent and sparse events.

For attributed dynamic graphs, we can also derive the corresponding edge ID sequences for source node $u$ with respect to node $v$, denoted as $E_{u \to v} = f^e_{u,v}(T_u) \subseteq \mathbb{N}^N$. The function $f^e_{u,v}(T_u)$ is defined as:

$$f^e_{u,v}(t) = \begin{cases} e_t, & \text{if } (u, v, t) \text{ or } (v, u, t) \in \mathcal{G} \\ 0, & \text{otherwise} \end{cases} \tag{2}$$

where $e_t$ denote the edge ID at timestamp $t$. Similarly, we can obtain $T_v, S_{v \to u}$ and $E_{v \to u}$ for destination node $v$ in the same manner.

**Embedding Generation Module.** In embedding generation module comprises three components: a time series projector, a time encoder, an edge projection layer and co-occurrence frequency encoder. These components are responsible for generating the interaction embedding, time embedding, edge embedding and co-occurrence frequency embedding from the binary time series, interaction timestamps, the edge sequences and co-occurrence of neighboring nodes. The time-series projector is implemented as a multi-layer perceptron (MLP). For source node $u$, the projected interaction embedding is computed as $X^A_u = \text{MLP}(S_{u \to v}) \in \mathbb{R}^{N \times d_A}$, where $d_A$ denotes the dimensions of the projected embedding. To capture the temporal information of the evolving interaction patterns, we adopt a time embedding proposed by previous work (Cong et al., 2023). The $i$-th entry of the time embedding for source node $u$ is formulated as

$$\tilde{X}^T_u[i] = \sqrt{\frac{1}{d_T}}[\cos(w_1 \Delta t_i), \cos(w_2 \Delta t_i), \dots, \cos(w_{d_T} \Delta t_i)], \tag{3}$$

where $\Delta t_i = t_c - t_i$ is the time interval between the current timestamp $t_c$ and the $i$-th timestamp $t_i \in T_u$. $[w_1, w_2, \dots, w_{d_T}]$ are trainable parameters, and $d_T$ denotes the dimension of the time embedding. The projected time embedding is obtained via linear transformation: $X^T_u = \tilde{X}^T_u W_T \in \mathbb{R}^{N \times d_C}$, where $W_T \in \mathbb{R}^{d_T \times d_C}$ represents the weight matrix of the time embedding projector.

The projected edge embedding is computed as: $X^E_u = \tilde{X}^E_u W_E \in \mathbb{R}^{N \times d_C}$, where $\tilde{X}^E_u \in \mathbb{R}^{N \times d_E}$ denotes the raw edge embeddings, and $W_E \in \mathbb{R}^{d_E \times d_C}$ is the weight matrix of the edge embedding projector. $d_E$ represents the dimension of raw edge embeddings. To address the lack of neighboring structural information

in our time-series dynamic graph setting, we adopt the co-occurrence frequency mechanism (Yu et al., 2023) to capture common neighboring nodes shared by the source and destination nodes, thereby modeling the temporal correlations of the target node pair. Suppose the neighbors of nodes $u$ and $v$ be $N_u = \{a, a, b\}$ and $N_v = \{b, c, c, a\}$, respectively. The co-occurrence features of source node $u$ can be denoted as $C_u = [[2, 1], [2, 1], [1, 1]]$, where $[2, 1]$ denotes the occurrence frequency of neighbor node $a$ in $N_u$ and $N_v$, respectively. Then the co-occurrence frequency embeddings can be obtained by $\mathbf{X}_u^{CO} = C_u[:, 0]\mathbf{W}_{CO} + C_u^{\tau}[:, 1]\mathbf{W}_{CO}$, where $\mathbf{W}_{CO}$ are learnable parameters.

The joint embedding for source node $u$ is constructed by concatenating projected interaction, time. edge and co-occurrence frequency embeddings. For attributed dynamic graph with edge features, $X_u^J = X_u^A || X_u^T || X_u^E || X_u^{CO} \in \mathbb{R}^{N \times 4d_C}$ or $X_u^J = X_u^A || X_u^T || X_u^{CO} \in \mathbb{R}^{N \times 3d_C}$ for non-attributed dynamic graph without edge features. We apply a linear transformation on joint embedding to obtain the input embedding $X_u^H = X_u^J W_H \in \mathbb{R}^{N \times d_H}$, where $W_H \in \mathbb{R}^{d_J \times d_H}$ is the projection matrix (with $d_J = 4d_C$ for attributed or $3d_C$ for non-attributed) and $d_H$ denotes the hidden dimension of the subsequent model. The corresponding embeddings for destination node $v$, i.e., $X_v^A, X_v^T, X_v^E, X_v^C, X_v^J$ and $X_v^H$ are computed in the same manner.

**Cross-attention Module.** Inspired by the previous work, we introduce the cross-attention mechanism to model the temporal patterns of the time series. Specifically, we use a learnable latent token $Z^L \in \mathbb{R}^{1 \times d_H}$ as the query which interacts with the key and value representations derived from the time series embedding. This design allows the model to distill the most relevant temporal information. Compared to the previous method such as DyGFormer (Yu et al., 2023) that adopts self-attention , cross attention has a linear time complexity, enabling our model to efficiently capture long-range temporal dependencies. The time complexity of our method is $\mathcal{O}(Nd_H)$. Compared with transformer-based methods such as DyGFormer (Yu et al., 2023) and TCL (Wang et al., 2021a), whose self-attention mechanism incurs $O(N^2 d_H)$ time complexity over historical interaction sequences, our cross-attention module reduces the attention cost to $O(Nd_H)$ by using a single learnable query token to attend to the sequence. For the source node $u$, the processing pipeline in the cross-attention module is illustrated as follows:

$$
\begin{aligned}
Z_u^0 &= Z_u^L, \\
Q_u^{i-1} &= Z_u^{i-1}W_Q, \ K_u^{i-1} = Z_u^H W_K, \ V_u^{i-1} = Z_u^H W_V, \\
Z_u^{i-1} &= \text{cross-attention}(Q_u^{i-1}, K_u^{i-1}, V_u^{i-1}), \\
Z_u^i &= \text{LN}(\text{FFN}(Z_u^{i-1}) + Z_u^{i-1}), \quad (i = 1, 2, 3),
\end{aligned}
\tag{4}
$$

where LN denotes layer normalization. The final output embedding for source node $u$ is denoted as $Z_u^O = Z_u^3$. The final output embedding for destination node $u$ is obtained using the same process.

**Training.** To predict the likelihood of an interaction between the source node $u$ and the destination node $v$, we employ a multi-layer perceptron (MLP) predictor that takes their final output embeddings as input: $\tilde{p} = \text{MLP}(Z_u^O, Z_v^O)$. The model is trained using the binary cross-entropy loss $\mathcal{L} = -\frac{1}{M}\sum_{i=1}^{M}(p_i \log(\tilde{p}_i) + (1 - p_i)\log(1 - \tilde{p}_i))$, where $M$ denotes the number of training samples (including both positive and negative pairs), and $p_i \in \{0, 1\}$ denote the ground-truth label.

# 5 Experiments

## 5.1 Experimental Settings

In this section, we first present the details of our experimental setup. We then conduct extensive experiments across multiple benchmark datasets, comparing the performance of our proposed method with several strong baselines. Finally, we provide an in-depth analysis of our model through ablation studies to investigate the contributions of each components.

**Datasets and Baselines.** In our experiments, we follow the routine to evaluate our method with nine datasets (Poursafaei et al., 2022) (Wikipedia, Reddit, LastFM, Enron, Social Evo., UCI, Flights, UN Trade, and Contact), which cover diverse domains. We also adopt five benchmark datasets from the Temporal Graph Benchmark (TGB) (Huang et al., 2023), namely tgbl-uci, tgbl-enron, tgbl-wiki, tgbl-subreddit,

Table 1: Comparison of performance in terms of AP between our proposed method and baseline models in the transductive setting. Random, historical, and inductive sampling strategies are adopted. Each experiment is repeated five times. Bold and underlined values indicate the best and second-best results, respectively.

| | Datasets | JODIE | DyRep | TGN | CAWN | TGAT | EdgeBank | GraphMixer | TCL | DyGFormer | DyG-Mamba | FreeDyG | TSDyG |
|---|---|---|---|---|---|---|---|---|---|---|---|---|---|
| rnd | Wikipedia | 96.50±0.14 | 94.86±0.06 | 98.45±0.06 | 98.76±0.03 | 96.94±0.06 | 90.37±0.00 | 97.25±0.03 | 96.47±0.16 | 99.03±0.02 | 99.06±0.01 | **99.26±0.01** | 99.20±0.01 |
| | Reddit | 98.31±0.14 | 98.22±0.04 | 98.63±0.06 | 99.11±0.01 | 98.52±0.02 | 94.86±0.00 | 97.31±0.01 | 97.53±0.02 | 99.22±0.01 | 99.25±0.00 | **99.48±0.01** | 99.32±0.01 |
| | LastFM | 70.85±2.13 | 71.92±2.21 | 77.07±3.97 | 86.99±0.06 | 73.42±0.21 | 79.29±0.00 | 75.61±0.24 | 67.27±2.16 | 93.00±0.12 | **94.22±0.04** | 92.15±0.16 | 93.64±0.11 |
| | Enron | 84.77±0.30 | 82.38±3.36 | 86.53±1.11 | 89.56±0.09 | 71.12±0.97 | 83.53±0.00 | 82.25±0.16 | 79.70±0.71 | 92.47±0.12 | **93.22±0.03** | 92.51±0.05 | 92.56±0.18 |
| | Social Evo. | 89.89±0.55 | 88.87±0.30 | 93.57±0.17 | 84.96±0.09 | 93.16±0.17 | 74.95±0.00 | 93.37±0.01 | 93.13±0.16 | 94.73±0.01 | 94.75±0.01 | **94.91±0.01** | 94.77±0.03 |
| | UCI | 89.43±1.09 | 65.14±2.30 | 92.34±1.04 | 95.18±0.06 | 79.63±0.70 | 76.20±0.00 | 93.25±0.57 | 89.57±1.63 | 95.79±0.17 | **96.79±0.08** | 96.10±0.11 | 96.17±0.05 |
| | UN Trade | 64.94±0.31 | 63.21±0.93 | 65.03±1.37 | 65.39±0.12 | 61.47±0.18 | 60.41±0.00 | 62.61±0.27 | 62.21±0.03 | 66.46±1.29 | **68.55±0.16** | 52.33±0.30 | 67.99±0.12 |
| | Contact | 95.31±1.33 | 95.98±0.15 | 68.91±0.12 | 90.26±0.28 | 96.28±0.09 | 92.58±0.00 | 91.92±0.03 | 92.44±0.12 | 98.29±0.01 | 98.37±0.01 | 97.41±0.16 | **98.57±0.02** |
| | Avg. Rank | 8.25 | 9.25 | 6.75 | 6.38 | 8.50 | 10.00 | 8.25 | 9.63 | 3.63 | **1.88** | 3.63 | **1.88** |
| his | Wikipedia | 83.01±0.66 | 79.93±0.56 | 86.86±0.33 | 71.21±1.67 | 87.38±0.22 | 73.35±0.00 | 90.90±0.10 | 89.05±0.39 | 82.23±2.54 | 82.12±1.22 | **91.59±0.57** | 87.52±0.36 |
| | Reddit | 80.03±0.36 | 79.83±0.31 | 81.22±0.61 | 80.82±0.45 | 79.55±0.20 | 73.59±0.00 | 78.44±0.18 | 77.14±0.16 | 81.57±0.67 | 81.16±0.11 | 85.06±1.01 | **85.18±0.47** |
| | LastFM | 74.35±3.81 | 74.92±2.46 | 76.87±4.64 | 69.86±0.43 | 71.59±0.24 | 73.03±0.00 | 72.47±0.49 | 59.30±2.31 | 81.57±0.48 | **84.09±0.44** | 79.71±0.51 | 84.04±0.35 |
| | Enron | 69.85±2.70 | 71.19±2.76 | 73.91±1.76 | 64.73±0.36 | 64.07±1.05 | 76.53±0.00 | 77.98±0.92 | 70.66±0.39 | 75.63±0.73 | 77.41±1.13 | 78.87±0.82 | **80.26±0.66** |
| | Social Evo. | 87.44±6.78 | 93.29±0.43 | 94.45±0.56 | 85.53±0.38 | 95.01±0.44 | 80.57±0.00 | 94.93±0.31 | 94.74±0.31 | 97.38±0.14 | 96.59±0.28 | 97.29±0.23 | **97.40±0.26** |
| | UCI | 75.24±5.80 | 55.10±3.14 | 80.43±2.12 | 65.30±4.43 | 68.27±1.37 | 65.50±0.00 | 84.11±1.35 | 80.25±2.74 | 82.17±0.82 | 82.95±2.24 | **86.10±1.19** | 84.73±0.60 |
| | UN Trade | 61.39±1.83 | 59.19±1.07 | 58.44±5.51 | 55.71±0.38 | 55.74±0.91 | **81.32±0.00** | 57.05±1.22 | 55.90±1.17 | 64.41±1.40 | 65.19±0.19 | 49.61±0.55 | 71.80±0.10 |
| | Contact | 95.31±2.13 | 96.39±0.20 | 66.22±0.10 | 84.16±0.49 | 96.05±0.52 | 88.81±0.00 | 93.36±0.41 | 93.86±0.21 | 97.57±0.06 | 97.80±0.14 | 96.36±0.23 | **97.88±0.21** |
| | Avg. Rank | 7.62 | 7.88 | 6.88 | 10.50 | 8.25 | 8.62 | 6.25 | 8.75 | 4.25 | 4.00 | 3.75 | **1.75** |
| ind | Wikipedia | 75.65±0.79 | 70.21±1.58 | 85.62±0.44 | 74.06±2.62 | 87.00±0.16 | 80.63±0.00 | 88.59±0.17 | 86.76±0.72 | 78.29±5.38 | 84.64±0.77 | **90.05±0.79** | 87.23±0.27 |
| | Reddit | 86.98±0.16 | 86.30±0.26 | 88.10±0.24 | 91.67±0.24 | 89.59±0.24 | 85.48±0.00 | 85.26±0.11 | 87.45±0.29 | 91.11±0.40 | 91.89±0.42 | 90.74±0.17 | **92.47±0.73** |
| | LastFM | 62.67±4.49 | 64.41±2.70 | 65.95±5.98 | 67.48±0.77 | 71.13±0.17 | **75.49±0.00** | 68.12±0.33 | 58.21±0.89 | 73.97±0.50 | 74.76±0.40 | 72.19±0.24 | 71.69±1.06 |
| | Enron | 68.96±0.98 | 67.79±1.53 | 70.89±2.72 | 75.15±0.58 | 63.94±1.36 | 73.89±0.00 | 75.01±0.79 | 71.29±0.32 | 77.41±0.89 | 79.90±0.90 | 77.81±0.65 | **80.51±0.12** |
| | Social Evo. | 89.82±4.11 | 93.28±0.48 | 95.13±0.56 | 88.32±0.27 | 94.84±0.44 | 83.69±0.00 | 94.72±0.33 | 94.90±0.36 | **97.68±0.10** | 96.91±0.24 | 97.57±0.65 | 97.65±0.42 |
| | UCI | 65.99±1.40 | 54.79±1.76 | 70.94±0.71 | 64.61±0.48 | 68.67±0.84 | 57.43±0.00 | **80.10±0.51** | 76.01±1.11 | 72.25±1.71 | 73.71±3.88 | 82.35±0.73 | 77.37±0.29 |
| | UN Trade | 60.42±1.48 | 60.19±1.24 | 61.04±6.01 | 62.54±0.67 | 60.61±1.24 | **72.97±0.00** | 60.15±1.29 | 61.06±1.74 | 55.79±1.02 | 61.88±1.46 | 49.64±0.50 | 72.83±0.38 |
| | Contact | 93.43±1.78 | 94.18±0.10 | 90.18±3.28 | 89.31±0.27 | 94.35±0.48 | 85.20±0.00 | 90.87±0.35 | 91.35±0.21 | 94.75±0.28 | 94.57±0.22 | 94.60±0.83 | **96.06±0.26** |
| | Avg. Rank | 9.25 | 9.88 | 7.38 | 7.75 | 6.88 | 7.88 | 7.00 | 7.00 | 5.00 | 3.75 | 4.00 | **2.25** |

and tgbl-lastfm, which span a diverse range of domains. Additional details about the datasets can be found in Appendix Sec. A.1. To comprehensively evaluate our proposed method, we compare it against seven popular dynamic graph learning methods: JODIE (Kumar et al., 2019), TGN (Rossi et al., 2020), TGAT (Xu et al., 2020), GraphMixer (Cong et al., 2023), TCL (Wang et al., 2021a), DyGFormer (Yu et al., 2023),FreeDyG (Tian et al., 2023) RepeatMixer (Zou et al., 2024) and DyG-Mamba (Li et al., 2025).

Table 2: Comparison of performance in terms of AUC between our proposed method and baseline models in the transductive setting. Random, historical, and inductive sampling strategies are adopted. Each experiment is repeated five times. Bold and underlined values indicate the best and second-best results, respectively.

| | Datasets | JODIE | DyRep | TGAT | TGN | CAWN | EdgeBank | TCL | GraphMixer | DyGFormer | DyG-Mamba | FreeDyG | TSDyG |
|---|---|---|---|---|---|---|---|---|---|---|---|---|---|
| rnd | Wikipedia | 96.33±0.07 | 94.37±0.09 | 96.67±0.07 | 98.37±0.07 | 98.54±0.04 | 90.78±0.00 | 95.84±0.18 | 96.92±0.03 | 98.91±0.02 | 98.96±0.00 | 99.10±0.01 | **99.15±0.01** |
| | Reddit | 98.31±0.05 | 98.17±0.05 | 98.47±0.02 | 98.60±0.06 | 99.01±0.01 | 95.37±0.00 | 97.42±0.02 | 97.17±0.09 | 99.15±0.01 | 99.05±0.01 | 99.20±0.00 | **99.28±0.01** |
| | LastFM | 70.49±1.66 | 71.16±1.89 | 71.59±0.18 | 78.47±2.94 | 85.92±0.10 | 83.77±0.00 | 64.06±1.16 | 73.53±0.12 | 93.05±0.10 | **93.99±0.02** | 93.42±0.15 | 93.67±0.00 |
| | Enron | 87.96±0.52 | 84.89±3.00 | 68.89±1.10 | 88.32±0.99 | 90.45±0.14 | 87.05±0.00 | 75.74±0.72 | 84.38±0.21 | **93.33±0.13** | 93.03±0.06 | 93.01±0.11 | 93.21±0.34 |
| | Social Evo. | 92.05±0.46 | 90.76±0.21 | 94.76±0.16 | 95.39±0.17 | 87.34±0.08 | 81.60±0.00 | 94.84±0.17 | 95.23±0.07 | 96.30±0.01 | 96.39±0.01 | **96.59±0.04** | 96.40±0.02 |
| | UCI | 90.44±0.49 | 68.77±2.34 | 78.53±0.74 | 92.03±1.13 | 93.87±0.07 | 77.30±0.00 | 87.82±1.36 | 91.81±0.67 | 94.49±0.26 | **96.50±0.06** | 95.00±0.21 | 94.93±0.03 |
| | UN Trade | 69.62±0.44 | 67.44±0.83 | 64.01±0.12 | 69.10±1.67 | 68.54±0.18 | 66.75±0.00 | 64.72±0.05 | 65.52±0.51 | 70.20±1.44 | 72.25±0.07 | 54.04±0.71 | **72.48±0.06** |
| | Contact | 96.66±0.89 | 96.48±0.14 | 96.95±0.08 | 97.54±0.35 | 89.99±0.34 | 94.34±0.00 | 94.15±0.09 | 93.94±0.02 | 98.53±0.01 | 98.58±0.01 | 97.99±0.10 | **98.83±0.02** |
| | Avg. Rank | 7.88 | 9.50 | 8.88 | 5.75 | 6.75 | 9.75 | 9.88 | 8.63 | 3.25 | 2.38 | 3.75 | **1.63** |
| his | Wikipedia | 80.77±0.73 | 77.74±0.33 | 82.87±0.22 | 82.74±0.32 | 67.84±0.64 | 77.27±0.00 | 85.76±0.46 | **87.68±0.17** | 78.80±1.95 | 78.93±1.42 | 82.28±0.30 | 82.44±0.46 |
| | Reddit | 80.52±0.32 | 80.15±0.18 | 79.33±0.16 | 81.11±0.19 | 80.27±0.30 | 78.58±0.00 | 76.49±0.16 | 77.80±0.12 | 80.54±0.29 | 80.96±0.20 | **85.92±0.10** | 81.81±0.42 |
| | LastFM | 75.23±2.36 | 74.65±1.98 | 64.27±0.26 | 77.97±3.04 | 67.88±0.24 | 78.09±0.00 | 47.24±3.13 | 64.21±0.73 | 78.78±0.35 | **80.88±0.52** | 73.53±0.12 | 79.87±0.37 |
| | Enron | 75.39±2.37 | 74.69±3.55 | 61.85±1.43 | 77.09±2.22 | 65.10±0.34 | **79.59±0.04** | 67.95±0.88 | 75.27±1.14 | 76.55±0.52 | 78.09±0.65 | 75.74±0.72 | 78.61±0.18 |
| | Social Evo. | 90.06±3.15 | 93.12±0.34 | 93.08±0.59 | 94.71±0.53 | 87.43±0.15 | 85.81±0.00 | 93.44±0.68 | 94.39±0.31 | 97.28±0.07 | 96.58±0.24 | 97.12±0.22 | **97.30±0.23** |
| | UCI | 78.64±3.50 | 57.91±3.12 | 58.89±1.57 | 77.25±2.68 | 78.56±0.15 | 69.56±0.00 | 72.25±3.46 | 77.54±2.02 | 76.97±0.24 | 77.35±1.25 | **80.38±0.26** | 77.39±0.18 |
| | UN Trade | 68.92±1.40 | 64.36±1.40 | 60.37±0.68 | 63.93±5.41 | 63.09±0.74 | **86.61±0.00** | 61.43±1.04 | 63.20±1.54 | 73.86±1.13 | 75.24±0.16 | 49.15±1,21 | 80.13±0.30 |
| | Contact | 96.35±0.92 | 96.00±0.23 | 95.39±0.43 | 93.76±1.29 | 83.06±0.32 | 92.17±0.00 | 93.34±0.19 | 93.14±0.34 | 97.17±0.05 | 97.43±0.11 | 95.88±0.13 | **97.57±0.28** |
| | Avg. Rank | 5.88 | 8.12 | 9.00 | 5.38 | 8.25 | 7.50 | 8.88 | 7.25 | 4.25 | 3.88 | 5.38 | **2.50** |
| ind | Wikipedia | 70.96±0.78 | 67.36±0.96 | 81.93±0.22 | 80.97±0.31 | 70.95±0.95 | 81.73±0.00 | 82.19±0.48 | **84.28±0.30** | 75.09±3.70 | 78.69±2.23 | 82.74±0.32 | 82.82±0.70 |
| | Reddit | 83.51±0.15 | 82.90±0.31 | 87.13±0.20 | 84.56±0.24 | **88.04±0.29** | 85.93±0.00 | 84.67±0.29 | 82.21±0.13 | 81.62±0.51 | 87.22±0.52 | 84.36±0.21 | 87.35±0.49 |
| | LastFM | 61.32±3.49 | 62.15±2.12 | 63.99±0.21 | 65.46±4.27 | 67.92±0.44 | 77.37±0.00 | 46.93±2.59 | 64.12±0.70 | 71.20±0.53 | 73.53±0.13 | **72.30±0.59** | 64.85±0.60 |
| | Enron | 70.92±1.05 | 68.73±1.34 | 60.45±2.12 | 71.34±2.46 | 75.17±0.50 | 75.00±0.00 | 67.64±0.86 | 71.53±0.85 | 74.07±0.64 | **77.79±0.54** | 77.21±0.61 | 77.66±0.07 |
| | Social Evo. | 90.01±3.19 | 93.07±0.38 | 92.94±0.61 | 93.08±0.56 | 89.93±0.15 | 87.88±0.00 | 93.44±0.72 | 94.22±0.32 | 97.48±0.06 | 96.78±0.21 | **98.47±0.20** | 97.51±0.45 |
| | UCI | 64.14±1.26 | 54.25±2.01 | 60.80±1.01 | 64.11±1.04 | 58.06±0.26 | 58.03±0.00 | 70.05±1.86 | 74.59±0.74 | 65.96±1.18 | 67.36±2.58 | **75.39±0.67** | 70.31±0.64 |
| | UN Trade | 66.82±1.27 | 65.60±1.28 | 66.13±0.78 | 66.37±5.39 | 71.73±0.74 | 74.20±0.00 | 67.80±1.21 | 66.53±1.22 | 62.56±1.51 | 70.82±1.14 | 49.21±1.12 | **78.57±0.60** |
| | Contact | 94.47±1.08 | 94.23±0.18 | 94.10±0.41 | 91.64±1.72 | 87.68±0.24 | 85.87±0.00 | 91.23±0.19 | 90.96±0.27 | 95.01±0.15 | 94.94±0.21 | 94.63±0.41 | **95.63±0.28** |
| | Avg. Rank | 8.25 | 9.62 | 7.88 | 7.12 | 7.00 | 6.75 | 7.25 | 6.88 | 6.50 | 4.00 | 4.25 | **2.50** |

**Evaluation Task and Metics.** In our experiments, we primarily focus on the future link prediction task, which is consistent with prior works. This task aims to predict the probability of an interaction occurring between two given nodes at a specific timestamp. It can be evaluated under two settings: the transductive setting, where all nodes are observed during training, and the inductive setting, where some nodes are unseen during training. In the evaluation, the random, historical and inductive negative sampling are adopted to

Table 3: Comparison of link prediction performance between our proposed method and baselines. Each experiment is repeated five times. Bold values indicate the best results.

| Category | Methods | tgbl-uci | | tgbl-enron | | tgbl-wiki | | tgbl-subreddit | | tgbl-lastfm | |
|---|---|---|---|---|---|---|---|---|---|---|---|
| | | MRR@20 | MRR@50 | MRR@20 | MRR@50 | MRR@20 | MRR@50 | MRR@20 | MRR@50 | MRR@20 | MRR@50 |
| Dynamic graph | JODIE | $63.45_{\pm 0.14}$ | $47.72_{\pm 0.38}$ | $43.78_{\pm 0.06}$ | $28.21_{\pm 0.20}$ | $80.84_{\pm 0.75}$ | $73.02_{\pm 1.22}$ | $88.25_{\pm 0.45}$ | $81.83_{\pm 0.62}$ | $40.81_{\pm 0.09}$ | $29.38_{\pm 0.07}$ |
| Dynamic graph | TGN | $63.97_{\pm 0.52}$ | $47.92_{\pm 0.64}$ | $27.02_{\pm 1.15}$ | $15.18_{\pm 1.71}$ | $88.13_{\pm 0.49}$ | $83.35_{\pm 0.73}$ | $89.57_{\pm 0.03}$ | $83.68_{\pm 0.11}$ | $46.11_{\pm 1.87}$ | $33.45_{\pm 1.97}$ |
| Dynamic graph | TGAT | $69.36_{\pm 0.38}$ | $52.88_{\pm 0.58}$ | $38.77_{\pm 0.08}$ | $23.23_{\pm 0.16}$ | $81.44_{\pm 0.61}$ | $74.77_{\pm 0.89}$ | $87.78_{\pm 0.07}$ | $81.34_{\pm 0.08}$ | $39.79_{\pm 0.17}$ | $30.57_{\pm 0.07}$ |
| Dynamic graph | GraphMixer | $72.40_{\pm 0.65}$ | $61.96_{\pm 0.05}$ | $53.14_{\pm 0.17}$ | $38.13_{\pm 0.14}$ | $83.46_{\pm 0.05}$ | $76.81_{\pm 0.34}$ | $86.80_{\pm 0.07}$ | $79.10_{\pm 0.10}$ | $45.70_{\pm 0.03}$ | $34.19_{\pm 0.24}$ |
| Dynamic graph | TCL | $63.20_{\pm 0.05}$ | $49.53_{\pm 0.03}$ | $40.43_{\pm 0.62}$ | $25.09_{\pm 0.49}$ | $86.48_{\pm 0.21}$ | $83.47_{\pm 0.28}$ | $87.49_{\pm 0.03}$ | $81.71_{\pm 0.06}$ | $48.45_{\pm 0.13}$ | $40.14_{\pm 0.14}$ |
| Dynamic graph | DyGFormer | $76.46_{\pm 0.04}$ | $69.89_{\pm 0.11}$ | $78.43_{\pm 0.22}$ | $69.90_{\pm 0.37}$ | $92.16_{\pm 0.11}$ | $89.95_{\pm 0.10}$ | $93.94_{\pm 0.03}$ | $91.06_{\pm 0.02}$ | $64.83_{\pm 0.01}$ | $55.01_{\pm 0.22}$ |
| Dynamic graph | FreeDyG | $80.46_{\pm 0.86}$ | $75.45_{\pm 0.97}$ | $77.86_{\pm 0.11}$ | $67.56_{\pm 0.81}$ | $93.56_{\pm 0.08}$ | $91.38_{\pm 0.02}$ | $93.64_{\pm 0.04}$ | $90.51_{\pm 0.01}$ | $64.16_{\pm 0.05}$ | $54.75_{\pm 0.06}$ |
| Dynamic graph | RepeatMixer | $79.82_{\pm 0.29}$ | $72.51_{\pm 0.23}$ | $79.42_{\pm 0.11}$ | $70.12_{\pm 0.19}$ | $92.84_{\pm 0.24}$ | $90.42_{\pm 0.52}$ | $\mathbf{94.51}_{\pm 0.12}$ | $\mathbf{92.17}_{\pm 0.15}$ | $70.66_{\pm 0.15}$ | $57.73_{\pm 0.12}$ |
| Time Series | BiLSTM | $70.85_{\pm 1.14}$ | $66.91_{\pm 1.12}$ | $77.27_{\pm 1.19}$ | $65.50_{\pm 1.26}$ | $88.63_{\pm 0.32}$ | $87.02_{\pm 0.53}$ | $83.36_{\pm 1.06}$ | $80.63_{\pm 1.12}$ | $70.10_{\pm 1.04}$ | $61.20_{\pm 1.25}$ |
| Time Series | iTransformer | $78.09_{\pm 0.12}$ | $71.80_{\pm 0.11}$ | $75.49_{\pm 0.14}$ | $64.00_{\pm 0.40}$ | $91.33_{\pm 0.28}$ | $88.66_{\pm 0.13}$ | $89.07_{\pm 0.10}$ | $84.07_{\pm 0.24}$ | $72.03_{\pm 0.20}$ | $61.98_{\pm 0.25}$ |
| Time Series | CATS | $72.06_{\pm 0.17}$ | $65.18_{\pm 0.23}$ | $79.84_{\pm 0.61}$ | $66.96_{\pm 0.77}$ | $90.21_{\pm 1.15}$ | $87.38_{\pm 0.68}$ | $78.58_{\pm 1.52}$ | $75.81_{\pm 1.56}$ | $72.91_{\pm 0.67}$ | $61.83_{\pm 0.36}$ |
| Joint | TSDyG (Ours) | $\mathbf{80.53}_{\pm 0.04}$ | $\mathbf{75.70}_{\pm 0.39}$ | $\mathbf{81.58}_{\pm 0.13}$ | $\mathbf{74.03}_{\pm 0.30}$ | $\mathbf{99.07}_{\pm 0.05}$ | $\mathbf{98.33}_{\pm 0.17}$ | $93.60_{\pm 0.05}$ | $91.02_{\pm 0.04}$ | $\mathbf{76.93}_{\pm 0.75}$ | $\mathbf{70.11}_{\pm 0.85}$ |

Table 4: Comparison of forecasting performance between our proposed method and baselines. Each experiment is repeated five times. Bold values indicate the best results.

| Category | Methods | tgbl-uci | | tgbl-enron | | tgbl-wiki | | tgbl-subreddit | | tgbl-lastfm | |
|---|---|---|---|---|---|---|---|---|---|---|---|
| | | MAE | MSE | MAE | MSE | MAE | MSE | MAE | MSE | MAE | MSE |
| Time Series | CATS | $0.338_{\pm 0.001}$ | $0.173_{\pm 0.001}$ | $0.341_{\pm 0.002}$ | $0.164_{\pm 0.001}$ | $0.135_{\pm 0.001}$ | $0.079_{\pm 0.001}$ | $0.482_{\pm 0.006}$ | $0.246_{\pm 0.001}$ | $0.391_{\pm 0.006}$ | $0.201_{\pm 0.002}$ |
| Joint | TSDyG | $\mathbf{0.267}_{\pm 0.002}$ | $\mathbf{0.136}_{\pm 0.001}$ | $\mathbf{0.242}_{\pm 0.003}$ | $\mathbf{0.125}_{\pm 0.004}$ | $\mathbf{0.131}_{\pm 0.002}$ | $\mathbf{0.071}_{\pm 0.002}$ | $\mathbf{0.121}_{\pm 0.002}$ | $\mathbf{0.060}_{\pm 0.001}$ | $\mathbf{0.308}_{\pm 0.009}$ | $\mathbf{0.160}_{\pm 0.003}$ |

form the negative examples. Average Precision (AP) and Area Under the Receiver Operating Characteristic Curve (AUC-ROC) are adopted as the evaluation metrics.

For the Temporal Graph Benchmark (TGB), we formulate link prediction as a ranking problem by sampling multiple negative examples for each positive interaction. For a positive example $(u, v, t)$, we fix the source node $u$ and the timestamp $t$, and sample multiple negative destination nodes $\tilde{v}$. These negative nodes are either randomly selected or historically sampled, where the latter are selected from nodes that previously interacted with $u$ but do not interact with u at the current timestamp $t$. We adopt the Mean Reciprocal Rank (MRR) as the evaluation metric, as suggested in TGB. MRR is calculated as the reciprocal of the rank of the true destination node among all candidate (true and negative) destination nodes.

**Model Configurations.** In our model, the dimensions of the interaction embeddings $d_A$, edge features $d_E$, and co-occurrence embeddings $d_{CO}$ are all set to 172. The dimension of the time embeddings $d_T$ is set to 100, while the dimension of the projected embeddings $d_C$ is set to 43. The hidden dimension $d_H$ is set to 172. The cross-attention module consists of three layers, and the time-series projector is implemented as a two-layer multilayer perceptron (MLP). For all the baselines, we follow their official implementation settings to ensure a fair comparison.

**Implementation Details.** we follow the routine to chronologically split each dataset with the ratio of 70%/15%/15% for training/validation/testing. The total training epoch is 100. We employ the Adam (Kingma & Ba, 2015) optimizer with a learning rate of 0.0001 and adopt an early stopping strategy with a patience of 20 epochs, selecting the model that performs best on the validation set for final evaluation. respectively. Each task is repeated five times, and all experiments are conducted on an NVIDIA RTX A5000. Additional implementation details can be found in Appendix Sec. A.2.

## 5.2  Main Results and Discussions

Table 1 and Table 2 report the performance of our method and the baseline models under different sampling strategies in the transductive setting. Overall, TSDyG achieves competitive performance across all sampling settings and shows clearer advantages under the historical and inductive settings. Under the random setting, TSDyG is generally competitive with strong baselines such as DyGFormer and DyG-Mamba, although some improvements are marginal and may fall within one standard deviation. In contrast, under historical and inductive sampling, the gains are more consistent and pronounced. For example, on the Enron dataset with historical sampling, our method achieves an average precision of 80.26%, and under inductive sampling it reaches 80.51%, both outperforming the strongest baselines by a clearer margin. These results suggest that our approach is particularly effective in settings that require stronger temporal generalization from historical

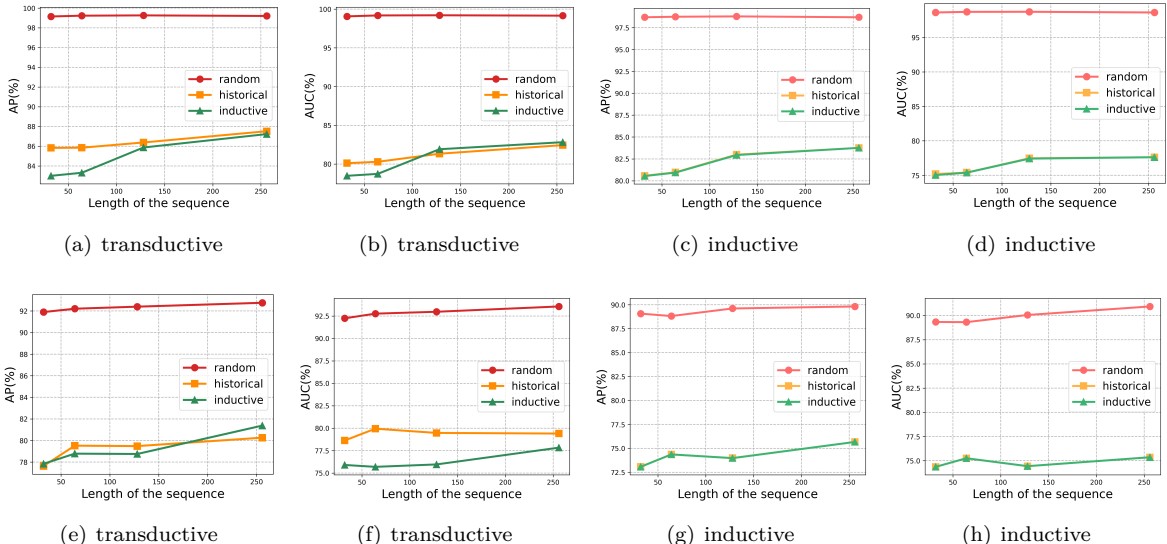

Figure 4: Performance of our method with varying sequence lengths from 32 to 256 under random, historical, and inductive sampling strategies. Evaluations are conducted on the Wikipedia dataset (first row) and the Enron dataset (second row).

interactions. Overall, the empirical findings indicate that our method can effectively capture regularities in evolving dynamic graphs. By explicitly focusing on the interactions between source–destination node pairs and distinguishing them from interactions involving the source node and other nodes, our approach constructs time series representations that preserve pair-specific temporal dynamics. Through this transformation of historical interactions, the model is able to learn how interaction patterns between target node pairs evolve as a function of time. Consequently, the proposed method can capture meaningful temporal regularities and leverage them to achieve strong performance on downstream prediction tasks. Additional results and discussions for the inductive setting can be found in Appendix Sec. A.3.

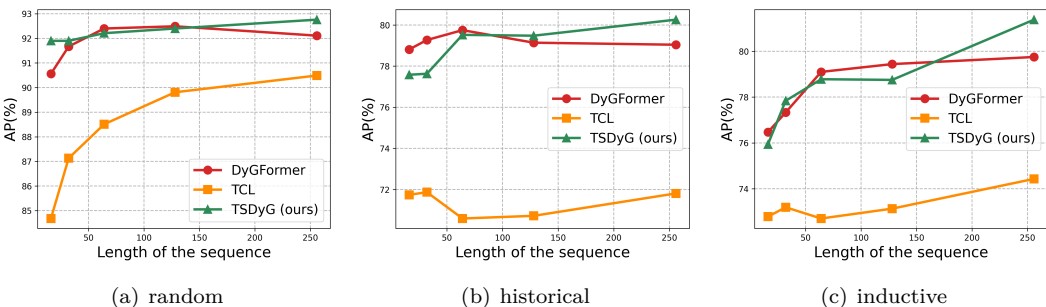

Figure 5: Performance comparison between our method and baseline models with varying sequence lengths from 16 to 256 under random, historical, and inductive sampling strategies in the transductive setting. Evaluations are conducted on the Enron dataset.

Table 3 reports the performance of our method and the baseline models on the Temporal Graph Benchmark (TGB) benchmarks (Huang et al., 2023). The results suggest that conventional time series methods such as Cross-Attention-only Time Series transformer (CATS) (Kim et al., 2024) and iTransformer (Liu et al., 2024) can be effectively adapted to the link prediction task in dynamic graphs. By converting dynamic graphs into time series using our proposed time series formulation module, these methods can achieve competitive performance. To further validate the effectiveness of time series-based approaches on binary data, we evaluate the performance of a traditional time series method such as CATS (Kim et al., 2024) and our proposed TSDyG model in a short-term forecasting setting. The fitness of each method on the binary time series data is assessed using Mean Squared Error (MSE) and Mean Absolute Error (MAE) between the

output probability and the ground-truth value (either 0 or 1). The results, presented in Table 4, suggest that traditional time series methods are indeed applicable to the short-term forecasting of our formulated binary time series data. However, the performance of time series methods is not consistently strong across all datasets. This variability can be attributed to two main reasons. First, traditional time series models are primarily designed for multivariate, continuous time series data, whereas the time series derived from dynamic graphs in our formulation are binary and discrete. As a result, the design of these methods may not be well-suited for our setting. Second, most time series methods are tailored for forecasting tasks (i.e., predicting future values in a continuous sequence), whereas our task involves predicting the probability of interaction between specific node pairs at a given timestamp. This task discrepancy limits the direct applicability and effectiveness of standard time series models in our setting.

The results in Table 3 show that TSDyG achieves strong and competitive performance across most tasks, and attains the best mean results on several datasets. For example, TSDyG attains average MRR@20 scores of 99.07% on tgbl-wiki and 76.93% on tgbl-lastfm (Huang et al., 2023), both of which exceed the performance of all baseline methods. The strong performance of TSDyG can be attributed to several key design choices that distinguish it from traditional dynamic graph models. Unlike prior approaches that aggregate temporal information from all neighboring nodes, our method reformulates historical interactions between target node pairs as binary time series by encoding interactions between the source and destination nodes as positive events ("1") and interactions between the source node and other nodes as negative events ("0"). This representation enables the model to focus exclusively on pair-specific interaction patterns. As a result, TSDyG can effectively capture temporal dependencies associated with target node pairs, including recurring and periodic interaction behaviors. Furthermore, by modeling interactions at the level of node pairs rather than individual nodes, TSDyG is better equipped to distinguish true interactions from negative or noisy interactions involving other nodes. In addition, the incorporation of a cross-attention mechanism facilitates the modeling of long-range temporal dependencies, allowing the model to comprehensively capture interaction dynamics over extended time horizons. This capability is particularly beneficial in large-scale dynamic graphs, where meaningful interaction patterns may span long periods of time. Besides, unlike time-series approaches, our model explicitly exploits the unique properties of dynamic graphs, including edge features and temporal structural information. These modalities, which are often neglected by traditional time-series methods, enhance the representation learning process and contribute to the competitive performance of our model on downstream tasks.

### 5.3 Analysis

#### 5.3.1 Sensitivity Analysis

In the sensitivity analysis, we examine how the performance of our proposed method varies with respect to the sequence length, that is, the number of historical interactions considered. To evaluate the impact of sequence length, we vary it across four values: 32, 64, 128, and 256, and conduct experiments on the Wikipedia and Enron dataset (Poursafaei et al., 2022). We evaluate the model under random, historical, and inductive sampling strategies, and report results in both transductive and inductive settings. The results are presented in Figure 4.

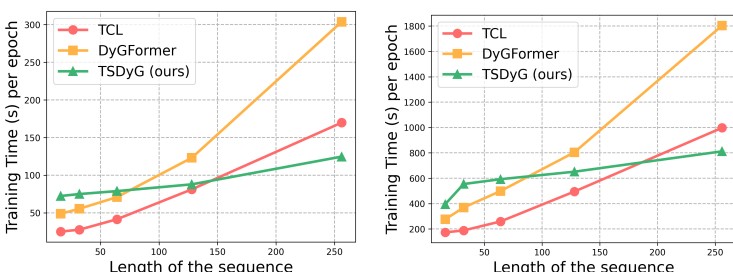

Figure 6: Training time per epoch comparison between our proposed TSDyG and baseline methods on the Enron (left) and Reddit (right) datasets.

Overall, the performance of our method remains stable across different sequence lengths under random sampling. In contrast, under historical and inductive sampling strategies, performance generally improves as the sequence length increases. This observation suggests that longer sequences provide more comprehensive information about interaction patterns between target node pairs, enabling the model to capture richer

temporal dependencies from historical interactions. Consequently, incorporating longer historical sequences leads to more accurate future interaction predictions.

### 5.3.2 Computational Cost Analysis

To evaluate the computational cost of our proposed model, we measure the training time of TSDyG and compare it with baseline methods, including DyGFormer (Yu et al., 2023) and TCL (Wang et al., 2021a). For a fair comparison, the hidden dimensionality is kept same across models and the batch size is fixed at 256. The sequence length is chosen from $\{16, 32, 64, 126, 256\}$. All experiments are conducted on NVIDIA A5000. Evaluations are performed on Enron and Reddit datasets (Poursafaei et al., 2022). The results illustrated in Figure 6 show that TSDyG incurs higher computational cost than the baselines when the sequence length is small (e.g., 16 or 32). This additional overhead mainly arises from transforming historical interactions into binary time-series representations, which is performed independently for each target node pair. However, as the sequence length increases, the attention modules begin to dominate the computational cost. Consequently, our method becomes significantly more efficient than the baselines, since the time complexity of TSDyG scales linearly with the sequence length, whereas most baseline methods exhibit quadratic complexity.

We also evaluate the performance of our proposed TSDyG and the baseline methods with varying sequence lengths on the Enron dataset (Poursafaei et al., 2022), as illustrated in Figure 5. Compared with DyG-Former (Yu et al., 2023), our model achieves slightly inferior performance when the sequence length is relatively small (e.g., less than 128). This is mainly because shorter interaction sequences provide limited temporal context, reducing the effectiveness of the proposed time-series modeling module. However, as the sequence length increases, TSDyG benefits more from the richer historical interaction information and is able to better capture long-range temporal patterns in node interactions. Consequently, the performance of our model improves consistently and eventually surpasses the baseline methods. These results further demonstrate the effectiveness of our method in modeling long-term temporal patterns in dynamic graphs. Additional results can be found in Appendix Sec. A.5.

### 5.4 Ablation Study

In the ablation study, we investigate the individual contributions of the key components in our proposed method, including the formulated binary time series, edge features, time embeddings, and co-occurrence encoding of neighboring nodes. All ablation experiments are conducted on the Wikipedia and Enron datasets (Poursafaei et al., 2022) under random, historical, and inductive sampling strategies.

We first evaluate the contribution of the formulated binary time series by removing the binary time series data from the model. The results are reported in Table 5. As shown, the model performance drops noticeably when the binary time series is excluded. For example, on the Enron dataset (Poursafaei et al., 2022), the average pre-

Table 5: Comparison of link prediction performance between our complete proposed method and our method without time series data, edge sequence, timestamps and co-occurrence. Each experiment is repeated five times.

| Datasets | | Wikipedia | | Enron | |
|---|---|---|---|---|---|
| | | AP | AUC | AP | AUC |
| rnd | w/o time series | 96.16±0.02 | 96.13±0.01 | 88.52±0.07 | 90.36±0.04 |
| | w/o edge sequence | 86.78±0.05 | 83.07±0.08 | 92.01±0.12 | 92.72±0.16 |
| | w/o timestamps | 87.88±0.03 | 84.75±0.05 | 92.06±0.20 | 91.38±0.25 |
| | w/o co-occurrence | 85.04±0.04 | 83.25±0.08 | 91.61±0.16 | 91.56±0.12 |
| | complete (ours) | **99.20±0.01** | **99.15±0.01** | **92.56±0.18** | **93.21±0.34** |
| his | w/o time series | 84.20±0.14 | 80.14±0.20 | 65.80±0.50 | 69.58±0.55 |
| | w/o edge sequence | 72.40±0.30 | 59.97±0.28 | 78.28±0.64 | 77.16±0.57 |
| | w/o timestamps | 62.79±0.23 | 58.18±0.33 | 77.05±0.83 | 78.32±0.93 |
| | w/o co-occurrence | 61.85±0.36 | 60.85±0.32 | 77.04±0.61 | 76.37±0.60 |
| | complete (ours) | **87.52±0.36** | **82.44±0.46** | **80.26±0.66** | **79.61±0.18** |
| ind | w/o time series | 82.11±0.24 | 79.58±0.27 | 71.66±0.21 | 72.41±0.20 |
| | w/o edge sequence | 71.38±0.10 | 58.30±0.19 | 78.56±0.14 | 76.82±0.20 |
| | w/o timestamps | 63.90±0.23 | 57.79±0.25 | 76.88±0.18 | 76.37±0.15 |
| | w/o co-occurrence | 56.14±0.18 | 53.10±0.16 | 77.04±0.24 | 75.87±0.32 |
| | complete (ours) | **87.23±0.27** | **82.82±0.70** | **80.51±0.12** | **77.66±0.07** |

cision decreases from 92.56% to 88.52% with random sampling strategy. This performance degradation indicates that the historical interaction patterns encoded in the binary time series are indispensable for accurate future interaction prediction. Next, we examine the effect of the time embedding by comparing model performance with and without temporal encoding. The results, presented in Table 5, show a significant performance drop when the time embedding is removed, particularly on the Wikipedia dataset (Poursafaei et al., 2022). This highlights the importance of temporal information in capturing the timing and order of

interactions. Without explicit time embeddings, the model struggles to distinguish target interactions from other historical interactions, leading to inferior predictive performance.

We also analyze the impact of edge features on attributed dynamic graphs. As shown in Table 5, removing edge features results in a noticeable decline in performance. These results demonstrate that edge attributes enhance the expressiveness of interaction representations and enable the model to better capture temporal dependencies across different node pairs. Finally, we evaluate the contribution of co-occurrence frequency encoding. The results indicate a consistent performance drop when co-occurrence information is removed. This suggests that the occurrence of shared neighboring nodes provides valuable structural signals that complement our model's limited ability to capture temporal structural information. By incorporating such co-occurrence patterns, the model can better characterize the evolving structural dependencies between nodes, thereby enhancing its representation learning capability and improving discrimination performance in link prediction tasks. Additional results can be found in Appendix Sec. A.4.

To isolate the contribution of the cross-attention module, we replace it with full self-attention and simple pooling while keeping all other model components unchanged. Experimental results illustrated in Table 6 on the Wikipedia and Enron datasets under random, historical, and inductive negative-sampling settings show that cross-attention achieves the best performance in most cases. Moreover, compared with full self-attention, it has lower computational complexity while maintaining or improving predictive performance. These results demonstrate that the proposed cross-attention module provides a favorable balance between effectiveness and efficiency.

Table 6: Average precision performance of TSDyG using cross-attention, full self-attention, and simple pooling on multiple benchmark datasets under random, historical, and inductive negative-sampling settings.

| Datasets | | Wikipedia | | Enron | |
|---|---|---|---|---|---|
| | | AP | AUC | AP | AUC |
| rnd | TSDyG (mean pooling) | 99.81±0.01 | 98.65±0.02 | 91.98±0.11 | 92.76±0.25 |
| | TSDyG (self-attention) | **99.20±0.01** | 99.13±0.01 | **92.84±0.21** | **93.36±0.30** |
| | TSDyG (cross-attention) | **99.20±0.01** | **99.15±0.01** | 92.56±0.18 | 93.21±0.34 |
| his | TSDyG (mean pooling) | 71.79±0.35 | 68.86±0.24 | 77.25±0.55 | 77.95±0.20 |
| | TSDyG (self-attention) | 83.13±0.40 | 79.32±0.33 | 79.57±0.75 | 78.47±0.30 |
| | TSDyG (cross-attention) | **87.52±0.36** | **82.44±0.46** | **80.26±0.66** | **78.61±0.18** |
| ind | TSDyG (mean pooling) | 77.07±0.30 | 72.59±0.35 | 75.13±0.08 | 73.30±0.06 |
| | TSDyG (self-attention) | 84.14±0.25 | 80.64±0.29 | 78.93±0.17 | 76.10±0.10 |
| | TSDyG (cross-attention) | **87.23±0.27** | **82.82±0.70** | **80.51±0.12** | **77.66±0.07** |

# 6 Simulation Study

To directly examine whether the performance advantage of TSDyG stems from its ability to capture recurring and periodic interaction patterns, we conduct a controlled simulation study on synthetic dynamic graphs. The central idea is to generate interaction sequences with known temporal regularities whose periodicity strength can be systematically varied.

We first construct a synthetic dynamic graph with $N = 100$ nodes and uniformly sample $K = 20$ target node pairs $(u_k, v_k)_{k=1}^{K}$ without replacement. Each target pair is assigned a phase offset $\phi_k \sim \text{Uniform}(0, 2\pi)$, such that different pairs reach their peak interaction probabilities at different times within the same period. We also generate low-probability background interactions among non-target pairs to prevent the models from solving the task by simply memorizing the identities of the active pairs.

Time is discretized into 200 ticks per 24-hour period, and the period length is therefore set to $P = 200$ ticks. For each target pair $(u_k, v_k)$, the probability of an interaction at time $t$ is defined as

$$p_k(t) = \text{clip}\left[\rho\left(\alpha\frac{1 + \sin\left(\frac{2\pi t}{P} + \phi_k\right)}{2} + (1-\alpha)\epsilon_{k,t}\right) + \eta_{k,t}, , 0, , 1\right], \tag{5}$$

where $\epsilon_{k,t} \sim \text{Uniform}(0,1)$ represents a temporally unstructured random component, $\eta_{k,t} \sim \mathcal{N}(0, 0.01)$ introduces observation noise, and $\rho = 1/6$ controls the overall interaction density. The periodic and random components have the same expected value, allowing us to vary the degree of temporal regularity while approximately preserving the average interaction rate. The parameter $\alpha \in [0,1]$ controls the strength of periodicity in the interaction patterns. Specifically, $\alpha = 0$ produces temporally random interactions with no periodic structure, $\alpha = 0.5$ represents moderate periodicity mixed with random variation, and $\alpha = 1.0$ produces a strong sinusoidal pattern corresponding to highly recurrent behavior.

To ensure that target-pair identities are not trivially predictive, we additionally generate non-periodic background interactions among non-target pairs. At each time tick, a small number of non-target pairs is sampled uniformly at random, and each selected pair generates an interaction with a constant probability $p_{\text{bg}} = \frac{\rho}{2}$. These background events introduce competing interactions while remaining independent of the periodicity parameter $\alpha$.

We simulate 30 periods, corresponding to a total of 6,000 time ticks. At each tick $t$, an interaction for every target pair is independently sampled according to $y_{k,t} \sim \text{Bernoulli}\big(p_k(t)\big)$. When $y_{k,t} = 1$, an interaction event $(u_k, v_k, t)$ is generated. All target-pair and background events are subsequently merged and sorted in chronological order. Under the selected density setting, each synthetic dataset contains approximately 10,000 target-pair interactions, in addition to the low-probability background events. Because clipping and Gaussian noise may slightly affect the realized interaction density, we also report the actual number of generated events for each value of $\alpha$.

We generate three synthetic datasets with $\alpha \in \{0.0, 0.5, 1.0\}$ while holding all other properties, including graph size, interaction density, period length, and noise level, constant so that periodicity strength is the primary varying factor. Each model is evaluated over three independent random seeds using the same chronological 70%/15%/15% training, validation, and test split adopted in the main experiments. As shown in Table 7, TSDyG consistently outperforms DyGFormer (Yu et al., 2023) across all periodicity levels and negative-sampling settings. More importantly, the performance gap generally widens as $\alpha$ increases. Under random negative sampling, the improvement grows from 0.45 AP points at $\alpha = 0$ to 1.06 points at $\alpha = 1.0$. Under historical sampling, it increases from 2.08 to 2.90 points. These controlled results provide direct evidence that the proposed pair-conditioned time-series formulation is particularly effective at exploiting recurring and periodic interaction patterns.

## 7 Limitation

Our proposed model primarily relies on the historical recurring interactions between node pairs to capture temporal dependencies. However, its effectiveness may be limited in dynamic graphs with low repeat interaction ratios, where sparse historical interactions provide insufficient information for modeling temporal patterns solely from individual node pairs. In such scenarios, incorporating additional contextual information from neighboring nodes and higher-order structural dependencies may be necessary to better characterize the underlying temporal dynamics.

Table 7: Average precision performance of DyGFormer and TSDyG on synthetic datasets with varying levels of temporal regularity under random, historical, and inductive negative-sampling settings. Each experiment is repeated three times.

| NSS | Model | $\alpha = 0$ | $\alpha = 0.5$ | $\alpha = 1.0$ |
|---|---|---|---|---|
| rnd | DyGFormer | 93.82±0.04 | 94.51±0.02 | 96.70±0.01 |
| | TSDyG (Ours) | **94.27±0.03** | **95.84±0.03** | **97.76±0.01** |
| his | DyGFormer | 83.26±0.45 | 84.90±0.36 | 85.96±0.27 |
| | TSDyG (Ours) | **85.34±0.37** | **87.63±0.41** | **88.86±0.25** |
| ind | DyGFormer | 81.46±0.50 | 83.08±0.33 | 84.48±0.23 |
| | TSDyG (Ours) | **82.58±0.53** | **84.87±0.39** | **87.03±0.18** |

## 8 Conclusion

We review prior work on discrete-time and continuous-time dynamic graph learning, which typically models dynamic graphs either as sequences of static graph snapshots or as streams of interaction events. Motivated by the observation that interactions between target node pairs often exhibit predictable temporal regularities, we propose a new formulation strategy that transforms dynamic interactions into time series representations. Based on this formulation, we introduce TSDyG, a novel model that explicitly focuses on interactions between target node pairs and captures the evolving patterns of dynamic graphs from a time-series perspective. Extensive experiments demonstrate that TSDyG effectively captures temporal dependencies and achieves strong performance across multiple benchmark datasets.

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

# A   Appendix

Table 8: Statistics of the datasets.

| Datasets | Domains | #Nodes | #Links | #N&L Feat | Bipartite | Duration | Unique Steps | Time Granularity |
|---|---|---|---|---|---|---|---|---|
| Wikipedia | Social | 9,227 | 157,474 | – & 172 | True | 1 month | 152,757 | Unix timestamps |
| Reddit | Social | 10,984 | 672,444 | – & 172 | True | 1 month | 669,065 | Unix timestamps |
| LastFM | Interaction | 1,980 | 1,293,103 | – & – | True | 1 month | 1,283,614 | Unix timestamps |
| Enron | Social | 184 | 125,235 | – & – | False | 3 years | 22,632 | Unix timestamps |
| Social Evo. | Proximity | 74 | 2,099,519 | – & 2 | False | 8 months | 565,932 | Unix timestamps |
| UCI | Social | 1,899 | 59,835 | – & – | False | 196 days | 58,911 | Unix timestamps |
| UN Trade | Economics | 255 | 507,497 | – & 1 | False | 32 years | 32 | years |
| Contact | Proximity | 692 | 2,426,279 | – & 1 | False | 1 month | 8,064 | 5 minutes |

Table 9: The details of Datasets.

| Dataset | Domain | #Nodes | #Edges | #Steps | Attributed |
|---|---|---|---|---|---|
| tgbl-uci | social | 3,212 | 59,835 | 58,911 | No |
| tgbl-enron | social | 365 | 125,235 | 22,632 | No |
| tgbl-wiki | rating | 9,227 | 157,474 | 152,757 | Yes |
| tgbl-subreddit | rating | 10,984 | 672,447 | 588,915 | Yes |
| tgbl-lastfm | recommendation | 1,980 | 1,293,103 | 1,283,614 | No |

## A.1   Details of Datasets

In our experiment we adopt multiple benchmarks covering different domains. The descriptions of these datasets are as follows:

**Wikipedia** is a bipartite temporal interaction dataset collected from user edits on Wikipedia pages, where edges represent interactions between users and pages with associated edge features and timestamps.

**Reddit** is a bipartite temporal interaction dataset derived from user posts on Reddit communities, where interactions between users and subreddits are associated with textual edge features and timestamps.

**LastFM** is a temporal music recommendation dataset that records user listening behaviors over time, where edges represent interactions between users and artists.

**Enron** is a temporal email communication network constructed from the Enron email corpus, where nodes correspond to employees and edges denote timestamped email exchanges.

**Social Evolution** is a proximity-based dynamic network collected from mobile device sensing data, where edges indicate physical proximity interactions between participants over time.

**UCI** is a temporal online communication dataset collected from the UC Irvine message platform, where edges represent timestamped message exchanges between users.

**UN Trade** is a temporal international trade network containing yearly trade relationships between countries, where edges represent trade interactions over time.

**Contact** is a temporal human contact network collected from wearable sensors, where edges indicate close-range face-to-face interactions between individuals at fine-grained temporal intervals.

**tgbl-uci** is a temporal social communication dataset collected from an online messaging platform at the University of California, Irvine, where edges represent timestamped message interactions between users.

**tgbl-enron** is a temporal email communication network derived from the Enron email corpus, where nodes correspond to employees and edges denote timestamped email exchanges.

**tgbl-wiki** is a temporal rating-style interaction dataset constructed from user editing activities on Wikipedia pages, where interactions are associated with edge features and timestamps.

**tgbl-subreddit** is a temporal interaction dataset collected from Reddit, where edges represent user interactions with subreddits over time and are accompanied by textual edge features.

**tgbl-lastfm** is a temporal recommendation dataset based on user music listening records from LastFM, where edges denote timestamped interactions between users and artists.

The details of the datasets are illustrated in Table 8 and Table 9.

## A.2 Experimental Setup

The implementation for the baselines are as follows:

**JODIE** (Kumar et al., 2019) models dynamic user-item interactions through recurrent memory updates and temporal embedding projection. Following the original implementation, we use one graph aggregation layer with 10 sampled temporal neighbors and a dropout rate of 0.1 under the recent-neighbor sampling strategy.

**TGN** (Rossi et al., 2020) extends memory-based dynamic graph learning with temporal message passing and node memory modules. We configure TGN with one aggregation layer, 10 sampled neighbors, a dropout rate of 0.1, and recent-neighbor sampling.

**TGAT** (Xu et al., 2020) employs temporal attention mechanisms with functional time encoding to model continuous-time dynamic graphs. We use two temporal attention layers with 20 sampled neighbors and a dropout rate of 0.1. Uniform neighbor sampling is adopted for Reddit, while recent-neighbor sampling is used for the remaining datasets.

**TCL** (Wang et al., 2021a) leverages transformer-based contrastive learning for temporal graph representation learning. We use two transformer layers with 20 sampled neighbors and a dropout rate of 0.1. Similar to TGAT, uniform sampling is adopted on Reddit and recent-neighbor sampling on the remaining datasets.

**GraphMixer** (Cong et al., 2023) models temporal interactions using MLP-Mixer style architectures for efficient dynamic graph learning. We use two mixer layers with a dropout rate of 0.5. The number of sampled neighbors is set to 10 on Reddit and 30 on the remaining datasets, together with recent-neighbor sampling.

**DyGFormer** (Yu et al., 2023) is a transformer-based dynamic graph model that encodes temporal interaction sequences through patch-wise self-attention. We use two transformer layers. For Reddit, the maximum input sequence length is set to 64 with patch size 2 and dropout 0.2; for the remaining datasets, the sequence length is set to 32 with patch size 1 and dropout 0.1.

**FreeDyG** (Tian et al., 2023) improves dynamic graph learning efficiency through frequency-domain temporal modeling and lightweight temporal aggregation. We follow the hyperparameter settings reported in the original paper for all datasets.

**RepeatMixer** (Zou et al., 2024) models repetitive temporal interaction patterns using mixer-based architectures and repeat-aware temporal representations. We adopt the same experimental settings as provided in the official implementation.

**DyG-Mamba** (Li et al., 2025) incorporates state-space modeling into dynamic graph learning to efficiently capture long-range temporal dependencies. We follow the default hyperparameter configurations suggested in the original work.

**Wikipedia** is a bipartite temporal interaction dataset collected from user edits on Wikipedia pages, where edges represent interactions between users and pages with associated edge features and timestamps.

In our TSDyG model, the cross-attention module consists of three layers, while the time-series projector is implemented as a two-layer multilayer perceptron (MLP). The dropout rate is fixed at 0.1 across all experiments. The sequence length is set to 64 for the UCI and tgbl-uci datasets, 256 for Wikipedia, Reddit, LastFM, Enron, UN Trade, tgbl-enron, tgbl-wiki, tgbl-subreddit, and tgbl-lastfm, and 512 for the Social Evolution and Contact datasets.

Table 10: Comparison of link prediction performance in terms of AP between our proposed method and baseline models in the inductive setting. Random, historical, and inductive sampling strategies are adopted. Each experiment is repeated five times. Bold and underlined values indicate the best and second-best results, respectively.

| | Datasets | JODIE | DyRep | TGAT | TGN | CAWN | TCL | GraphMixer | DyGFormer | DyG-Mamba | FreeDyG | TSDyG |
|---|---|---|---|---|---|---|---|---|---|---|---|---|
| rnd | Wikipedia | 94.82±0.20 | 92.43±0.37 | 96.22±0.07 | 97.83±0.04 | 98.24±0.03 | 96.22±0.17 | 96.65±0.02 | 98.59±0.03 | 98.66±0.02 | **98.97±0.01** | 98.69±0.03 |
| | Reddit | 96.50±0.13 | 96.09±0.11 | 97.09±0.04 | 97.50±0.07 | 98.62±0.01 | 94.09±0.07 | 95.26±0.02 | 98.84±0.02 | 98.91±0.01 | 98.90±0.01 | **98.92±0.00** |
| | LastFM | 81.61±3.82 | 83.02±1.48 | 78.63±0.31 | 81.45±4.29 | 89.42±0.07 | 73.53±1.66 | 82.11±0.42 | 94.23±0.09 | 95.16±0.05 | 94.89±0.01 | **95.70±0.25** |
| | Enron | 80.72±1.39 | 74.55±3.95 | 67.05±1.51 | 77.94±1.02 | 86.35±0.51 | 76.14±0.79 | 75.88±0.48 | 89.76±0.34 | **90.97±0.01** | 89.69±0.17 | 89.83±0.08 |
| | Social Evo. | 91.96±0.48 | 90.04±0.47 | 91.41±0.16 | 90.77±0.86 | 79.94±0.18 | 91.55±0.09 | 91.86±0.06 | 93.14±0.04 | 93.17±0.05 | **94.76±0.05** | 93.37±0.03 |
| | UCI | 79.86±1.48 | 57.48±1.87 | 79.54±0.48 | 88.12±2.05 | 92.73±0.06 | 87.36±2.03 | 91.19±0.42 | 94.54±0.12 | 93.38±0.21 | 94.65±0.10 | **94.75±0.02** |
| | UN Trade | 59.65±0.77 | 57.02±0.69 | 61.03±0.18 | 58.31±3.15 | 65.24±0.21 | 62.21±0.12 | 62.17±0.31 | 64.55±0.62 | 67.04±0.20 | 51.95±0.76 | **67.25±0.27** |
| | Contact | 94.34±1.45 | 92.18±0.41 | 95.87±0.11 | 93.82±0.99 | 89.55±0.30 | 91.11±0.12 | 90.59±0.05 | 98.03±0.02 | 98.10±0.02 | 97.13±0.21 | **98.24±0.03** |
| | Avg. Rank | 7.50 | 9.38 | 8.31 | 7.50 | 6.25 | 8.44 | 7.63 | 3.63 | 2.38 | 3.63 | **1.38** |
| his | Wikipedia | 68.69±0.39 | 62.18±1.27 | 84.17±0.22 | 81.76±0.32 | 67.27±1.63 | 82.20±2.18 | **87.60±0.30** | 71.42±4.43 | 73.68±4.06 | 82.78±0.30 | 83.76±0.06 |
| | Reddit | 62.34±0.54 | 61.60±0.72 | 63.47±0.36 | 64.85±0.85 | 63.67±0.41 | 60.83±0.25 | 64.50±0.26 | 65.37±0.60 | 66.74±0.13 | 66.02±0.41 | **69.47±0.52** |
| | LastFM | 70.39±4.31 | 71.45±1.76 | 76.27±0.25 | 66.65±6.11 | 71.33±0.47 | 65.78±0.65 | 76.42±0.22 | 76.35±0.52 | **79.22±0.33** | 77.28±0.21 | 77.68±0.86 |
| | Enron | 65.86±3.71 | 62.08±2.27 | 61.40±1.31 | 62.91±1.16 | 60.70±0.36 | 67.11±0.62 | 72.37±1.37 | 67.07±0.62 | 75.12±1.43 | 73.01±0.88 | **75.67±1.22** |
| | Social Evo. | 88.51±0.87 | 88.72±1.10 | 93.97±0.54 | 90.66±1.62 | 79.83±0.38 | 94.10±0.31 | 94.01±0.47 | 96.82±0.16 | 95.45±0.30 | 96.69±0.14 | **96.92±0.27** |
| | UCI | 63.11±2.27 | 52.47±2.06 | 70.52±0.93 | 70.78±0.78 | 64.54±0.47 | 76.71±1.00 | 81.66±0.49 | 72.13±1.87 | 73.65±3.70 | **82.35±0.39** | 77.17±0.67 |
| | UN Trade | 55.46±1.19 | 55.49±0.84 | 55.28±0.71 | 52.80±3.19 | 55.00±0.38 | 55.76±1.03 | 54.94±0.97 | 53.20±1.07 | **56.80±0.27** | 49.31±0.98 | 56.05±0.25 |
| | Contact | 90.42±2.34 | 89.22±0.66 | 94.15±0.45 | 88.13±1.50 | 74.20±0.80 | 90.44±0.17 | 89.91±0.36 | 93.56±0.52 | 93.66±0.56 | 93.38±0.19 | **94.84±0.40** |
| | Avg. Rank | 8.25 | 8.75 | 6.12 | 8.00 | 9.25 | 6.25 | 4.88 | 5.50 | 3.12 | 4.12 | **1.75** |
| ind | Wikipedia | 68.70±0.39 | 62.19±1.28 | 84.17±0.22 | 81.77±0.32 | 67.24±1.63 | 82.21±2.18 | **87.60±0.29** | 71.42±4.43 | 73.68±4.06 | 87.54±0.26 | 83.76±0.70 |
| | Reddit | 62.32±0.54 | 61.58±0.72 | 63.40±0.36 | 64.84±0.84 | 63.65±0.41 | 60.81±0.26 | 64.49±0.25 | 65.35±0.60 | 66.74±0.13 | 64.98±0.20 | **67.48±0.83** |
| | LastFM | 70.39±4.31 | 71.45±1.75 | 76.28±0.25 | 69.46±4.65 | 71.33±0.47 | 65.78±0.65 | 76.42±0.22 | 76.35±0.52 | **79.22±0.33** | 76.01±0.43 | 76.93±0.96 |
| | Enron | 65.86±3.71 | 62.08±2.27 | 61.40±1.30 | 62.90±1.16 | 60.72±0.36 | 67.11±0.62 | 72.37±1.38 | 67.07±0.62 | 75.12±1.43 | 72.85±0.81 | **75.67±1.10** |
| | Social Evo. | 88.51±0.87 | 88.72±1.10 | 93.97±0.54 | 90.65±1.62 | 79.83±0.39 | 94.10±0.32 | 94.01±0.47 | 96.82±0.17 | 95.45±0.30 | 96.81±0.12 | **96.92±0.38** |
| | UCI | 63.16±2.27 | 52.47±2.09 | 70.49±0.93 | 70.73±0.79 | 64.54±0.47 | 76.65±0.99 | 81.64±0.49 | 72.13±1.86 | 73.65±3.70 | **82.06±0.58** | 76.85±0.55 |
| | UN Trade | 55.43±1.20 | 55.42±0.87 | 55.58±0.68 | 52.80±3.24 | 49.97±0.38 | 55.66±0.98 | 54.88±1.01 | 52.56±1.70 | 56.80±0.27 | 49.31±0.98 | **58.09±0.25** |
| | Contact | 90.43±2.33 | 89.22±0.65 | 94.14±0.45 | 88.12±1.50 | 74.19±0.81 | 90.43±0.17 | 89.91±0.36 | 93.55±0.52 | 93.66±0.56 | 93.38±0.20 | **94.86±0.34** |
| | Avg. Rank | 8.19 | 9.00 | 5.88 | 7.75 | 9.62 | 6.31 | 4.62 | 5.25 | 3.25 | 4.38 | **1.75** |

Table 11: Comparison of link prediction performance in terms of AUC between our proposed method and baseline models in the inductive setting. Random, historical, and inductive sampling strategies are adopted. Each experiment is repeated five times. Bold and underlined values indicate the best and second-best results, respectively.

| | Datasets | JODIE | DyRep | TGAT | TGN | CAWN | TCL | GraphMixer | DyGFormer | DyG-Mamba | FreeDyG | TSDyG |
|---|---|---|---|---|---|---|---|---|---|---|---|---|
| rnd | Wikipedia | 94.33±0.27 | 91.49±0.45 | 95.90±0.09 | 97.72±0.03 | 98.03±0.04 | 95.57±0.20 | 96.30±0.04 | 98.48±0.03 | 98.55±0.01 | **99.01±0.02** | 98.64±0.01 |
| | Reddit | 96.52±0.13 | 96.05±0.12 | 96.98±0.04 | 97.39±0.07 | 98.42±0.02 | 93.80±0.07 | 94.97±0.05 | 98.71±0.01 | **98.91±0.01** | 98.84±0.01 | 98.88±0.01 |
| | LastFM | 81.13±3.39 | 82.24±1.51 | 76.99±0.29 | 82.61±3.15 | 87.82±0.12 | 70.84±0.85 | 80.37±0.18 | 94.08±0.08 | 94.82±0.01 | 94.32±0.03 | **94.86±0.06** |
| | Enron | 81.96±1.34 | 76.34±4.20 | 64.63±1.74 | 78.83±1.11 | 87.02±0.50 | 72.33±0.99 | 76.51±0.71 | **90.69±0.26** | 90.60±0.11 | 89.51±0.20 | 90.68±0.00 |
| | Social Evo. | 93.70±0.29 | 91.18±0.49 | 93.41±0.19 | 93.43±0.59 | 84.73±0.27 | 93.71±0.18 | 94.09±0.07 | 95.29±0.03 | 95.41±0.01 | **96.41±0.07** | 95.46±0.04 |
| | UCI | 78.80±0.94 | 58.08±1.81 | 77.64±0.38 | 86.68±2.29 | 90.40±0.11 | 84.49±1.82 | 89.30±0.57 | 92.63±0.13 | 92.05±0.23 | **93.01±0.08** | 92.80±0.03 |
| | UN Trade | 62.28±0.50 | 58.82±0.98 | 62.72±0.12 | 59.99±3.50 | 67.05±0.21 | 63.76±0.07 | 63.48±0.37 | 67.25±1.05 | 69.37±0.06 | 53.47±0.91 | **69.52±0.21** |
| | Contact | 95.37±0.92 | 91.89±0.38 | 96.53±0.10 | 94.84±0.75 | 89.07±0.34 | 93.05±0.09 | 92.83±0.05 | 98.30±0.02 | 98.33±0.01 | 97.75±0.13 | **98.36±0.05** |
| | Avg. Rank | 7.75 | 9.62 | 8.38 | 7.00 | 6.38 | 8.50 | 7.50 | 3.25 | 2.50 | 3.50 | **1.62** |
| his | Wikipedia | 61.86±0.53 | 57.54±1.09 | 78.38±0.20 | 75.75±0.29 | 62.04±0.65 | 79.79±0.96 | 82.87±0.21 | 68.33±2.82 | 69.73±0.48 | **82.08±0.32** | 77.62±0.46 |
| | Reddit | 61.69±0.39 | 60.45±0.37 | 64.43±0.27 | 64.55±0.50 | 64.94±0.21 | 61.43±0.26 | 64.27±0.13 | 64.81±0.25 | **67.82±0.30** | 66.29±0.31 | 66.55±0.53 |
| | LastFM | 68.44±3.26 | 68.79±1.08 | 69.89±0.28 | 66.99±5.62 | 67.69±0.24 | 55.88±1.85 | 70.07±0.20 | 70.73±0.37 | 72.52±0.31 | **72.63±0.16** | 70.42±0.43 |
| | Enron | 65.32±3.57 | 61.50±2.50 | 57.84±1.18 | 62.68±1.09 | 62.25±0.40 | 64.06±1.02 | 68.20±1.62 | 65.78±0.42 | 75.35±1.06 | 70.09±0.65 | **75.36±0.69** |
| | Social Evo. | 88.53±0.55 | 87.93±1.05 | 91.87±0.72 | 92.10±1.22 | 83.54±0.24 | 93.28±0.60 | 93.62±0.35 | 96.91±0.09 | 96.03±0.24 | 96.64±0.17 | **96.95±0.33** |
| | UCI | 60.24±1.94 | 51.25±2.37 | 62.32±1.18 | 62.69±0.90 | 56.59±0.10 | 70.46±1.94 | 75.98±0.84 | 65.55±1.01 | 66.93±2.56 | **76.01±0.75** | 69.94±0.52 |
| | UN Trade | 58.73±1.19 | 57.90±1.33 | 59.74±0.59 | 55.61±3.54 | 60.95±0.80 | 61.12±0.97 | 59.88±1.17 | 58.46±1.65 | 62.81±0.21 | 58.46±1.65 | **63.59±0.09** |
| | Contact | 90.80±1.18 | 88.88±0.68 | 93.76±0.41 | 88.84±1.39 | 74.79±0.37 | 90.37±0.16 | 90.04±0.29 | 94.14±0.26 | 94.18±0.36 | 93.85±0.22 | **94.78±0.32** |
| | Avg. Rank | 8.00 | 9.88 | 6.88 | 8.12 | 8.38 | 6.25 | 4.75 | 5.06 | 3.12 | 3.19 | **2.38** |
| ind | Wikipedia | 61.87±0.53 | 57.54±1.09 | 78.38±0.20 | 75.76±0.29 | 62.02±0.65 | 79.79±0.96 | 82.88±0.21 | 68.33±2.82 | 69.73±0.48 | **83.17±0.31** | 77.62±0.36 |
| | Reddit | 61.69±0.39 | 60.44±0.37 | 64.39±0.27 | 64.55±0.50 | 64.91±0.21 | 61.36±0.26 | 64.27±0.13 | 64.81±0.30 | **66.78±0.36** | 64.51±0.19 | 66.55±0.53 |
| | LastFM | 68.44±3.26 | 68.79±1.08 | 69.89±0.28 | 66.99±5.61 | 67.68±0.24 | 55.88±1.85 | 70.07±0.20 | 70.73±0.37 | **72.52±0.31** | 71.42±0.33 | 70.12±0.43 |
| | Enron | 65.32±3.57 | 61.50±2.50 | 57.83±1.18 | 62.68±1.09 | 62.27±0.40 | 64.06±1.02 | 68.19±1.63 | 65.79±0.42 | 75.35±1.06 | 68.79±0.91 | **75.43±0.69** |
| | Social Evo. | 88.53±0.55 | 87.93±1.05 | 91.88±0.72 | 92.10±1.22 | 83.54±0.24 | 93.28±0.60 | 93.62±0.35 | 96.91±0.09 | 96.03±0.24 | 96.79±0.17 | **96.95±0.40** |
| | UCI | 60.27±1.94 | 51.26±2.40 | 62.29±1.17 | 62.66±0.91 | 56.59±0.10 | 70.42±0.99 | **75.97±0.85** | 65.58±1.00 | 66.93±2.56 | 73.41±0.88 | 69.81±0.25 |
| | UN Trade | 58.71±1.20 | 57.87±1.36 | 59.98±0.59 | 55.62±3.59 | 48.97±0.38 | 56.61±0.98 | 54.88±1.01 | 52.56±1.70 | 56.80±0.27 | 48.55±2.06 | **63.62±0.09** |
| | Contact | 90.80±1.18 | 88.87±0.67 | 93.76±0.40 | 88.85±1.39 | 74.79±0.38 | 90.37±0.16 | 90.04±0.29 | 94.14±0.26 | 94.18±0.36 | 93.84±0.23 | **94.75±0.26** |
| | Avg. Rank | 7.50 | 9.12 | 6.38 | 7.50 | 9.00 | 6.62 | 5.12 | 5.00 | 3.38 | 4.00 | **2.38** |

## A.3 Main Results

The performance of our method and the baseline models in the inductive setting is presented in Table 10 and Table 11, respectively. Overall, the experimental results demonstrate that TSDyG achieves highly competitive performance across different datasets and negative sampling strategies, with the clearest improvements observed under the historical and inductive settings. In particular, TSDyG attains the best average ranking under almost all settings in both AP and AUC metrics, highlighting its strong generalization capability in the inductive dynamic link prediction tasks.

Table 12: Comparison of link prediction performance between our complete proposed method and our method without time series data, edge sequence, timestamps and co-occurrence in the inductive setting. Each experiment is repeated 5 times. Bold values indicate the best results.

| Datasets | | Wikipedia | | Enron | |
|---|---|---|---|---|---|
| | | AP | AUC | AP | AUC |
| rnd | w/o time series | 98.60±0.03 | 98.59±0.02 | 80.53±0.04 | 84.49±0.03 |
| | w/o edge sequence | 87.23±0.04 | 83.65±0.10 | 89.21±0.10 | 90.14±0.15 |
| | w/o timestamps | 88.88±0.03 | 86.99±0.07 | 89.20±0.16 | 90.30±0.27 |
| | w/o co-occurrence | 85.71±0.03 | 84.15±0.07 | 89.26±0.16 | 89.63±0.14 |
| | complete (ours) | **98.69±0.03** | **98.64±0.01** | **89.83±0.08** | **90.68±0.06** |
| his | w/o time series | 77.16±0.18 | 73.23±0.23 | 63.64±0.54 | 67.35±0.52 |
| | w/o edge sequence | 74.13±0.26 | 63.35±0.32 | 74.70±0.60 | 74.91±0.55 |
| | w/o timestamps | 63.28±0.20 | 59.54±0.36 | 73.52±0.78 | 74.87±0.90 |
| | w/o co-occurrence | 61.26±0.35 | 57.43±0.36 | 73.60±0.64 | 73.44±0.63 |
| | complete (ours) | **83.76±0.06** | **77.62±0.46** | **75.67±1.22** | **75.36±0.69** |
| ind | w/o time series | 77.17±0.21 | 73.23±0.24 | 63.64±0.23 | 67.35±0.24 |
| | w/o edge sequence | 74.13±0.13 | 63.35±0.19 | 74.70±0.14 | 74.92±0.22 |
| | w/o timestamps | 63.28±0.22 | 59.54±0.28 | 73.52±0.19 | 74.87±0.14 |
| | w/o co-occurrence | 61.26±0.20 | 57.43±0.13 | 73.61±0.23 | 73.45±0.30 |
| | complete (ours) | **83.76±0.70** | **77.62±0.36** | **75.67±1.10** | **75.43±0.69** |

Under the random negative sampling setting, TSDyG remains competitive and achieves the highest mean performance on several datasets, although in some cases the gains over the strongest baselines are relatively small and fall within one standard deviation, especially on Wikipedia, Reddit, LastFM, UCI, UN Trade, and Contact. These results indicate that the proposed temporal sequence modeling framework effectively captures the long-range temporal dependencies and repetitive interaction patterns within dynamic graphs. Although DyG-Mamba and DyGFormer achieve competitive performance on several datasets such as Enron and Contact, TSDyG still obtains the best overall average ranking. Under the more challenging historical and inductive negative sampling settings, the performance gap between TSDyG and existing methods becomes more consistent and more pronounced. Historical and inductive negative samples are more difficult since they require the model to distinguish positive interactions from previously observed or structurally similar node pairs. The competitive performance of TSDyG under these settings suggests that modeling temporal interaction sequences as time series effectively enhances the discriminative capability of dynamic graph representations.

Furthermore, compared with transformer-based baselines such as DyGFormer and mixer-based approaches such as GraphMixer, TSDyG achieves consistently strong performance while maintaining better scalability for long interaction sequences. This demonstrates the effectiveness of integrating temporal sequence modeling with dynamic graph structural information for temporal link prediction. In summary, the results validate that TSDyG provides a robust and effective framework for dynamic graph learning, particularly in challenging inductive scenarios where capturing long-term temporal dependencies and repetitive interaction behaviors is essential.

## A.4 Ablation Study

To investigate the effectiveness of different components in our proposed TSDyG framework, we conduct ablation studies by progressively removing the time-series modeling module, edge sequence information, timestamp encoding, and co-occurrence features. The experimental results are reported in Table 12.

Overall, the complete TSDyG model consistently achieves the best performance across all sampling strategies and evaluation metrics, demonstrating the importance of jointly modeling temporal interaction sequences and structural information in dynamic graphs. Removing the time-series modeling module leads to noticeable performance degradation, particularly under the historical and inductive settings. This observation highlights that explicitly modeling temporal interaction sequences as time series plays a critical role in capturing long-range temporal dependencies and repetitive interaction behaviors. Eliminating edge sequence information or timestamp encoding also substantially reduces performance, indicating that both interaction semantics and temporal ordering are essential for effective temporal representation learning. In particular, timestamp

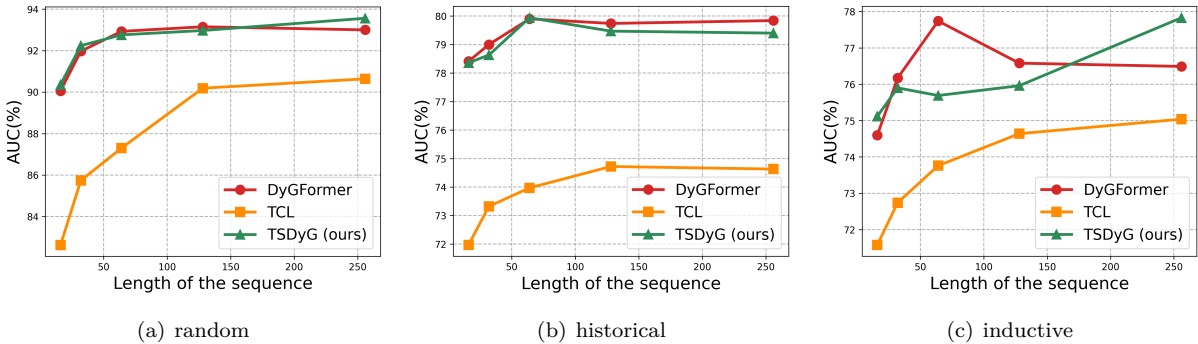

Figure 7: Performance of our method with different sequence lengths ranging from 32 to 256 under random, historical, and inductive sampling strategies. The evaluations are performed on Wikipedia.

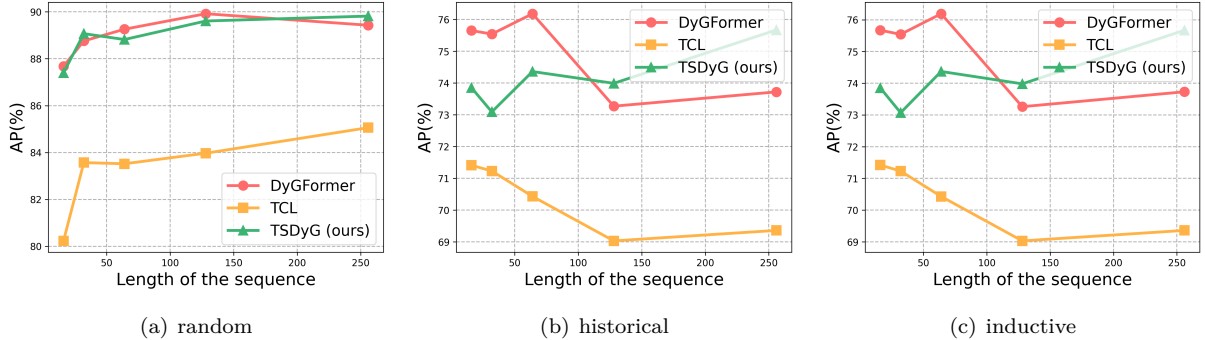

Figure 8: Performance of our method with different sequence lengths ranging from 32 to 256 under random, historical, and inductive sampling strategies. The evaluations are performed on Wikipedia.

information helps the model better characterize temporal evolution patterns, while edge sequences preserve informative interaction contexts between node pairs.

Furthermore, removing the co-occurrence module consistently decreases performance on both Wikipedia and Enron datasets. This result suggests that shared neighboring nodes provide valuable temporal structural signals that complement pairwise interaction modeling and improve the model's discriminative capability. In summary, the ablation results verify that each component of TSDyG contributes positively to the overall performance, and the combination of time-series modeling, temporal encoding, edge interaction sequences, and co-occurrence information is crucial for achieving strong dynamic link prediction performance.

### A.5  Analysis

We further investigate the impact of sequence length on model performance under random, historical, and inductive sampling strategies, as illustrated in Figure 7, Figure 8, and Figure 9. Overall, the performance of TSDyG consistently improves as the sequence length increases, especially under the historical and inductive settings. This observation suggests that longer historical interaction sequences provide richer temporal dependency patterns that are beneficial for dynamic link prediction. Compared with DyGFormer and TCL, our proposed TSDyG demonstrates stronger scalability with respect to sequence length. While DyGFormer tends to saturate or fluctuate when the sequence length becomes large, TSDyG continues to benefit from additional temporal context and achieves stronger or competitive performance under most settings, especially under the historical and inductive settings.

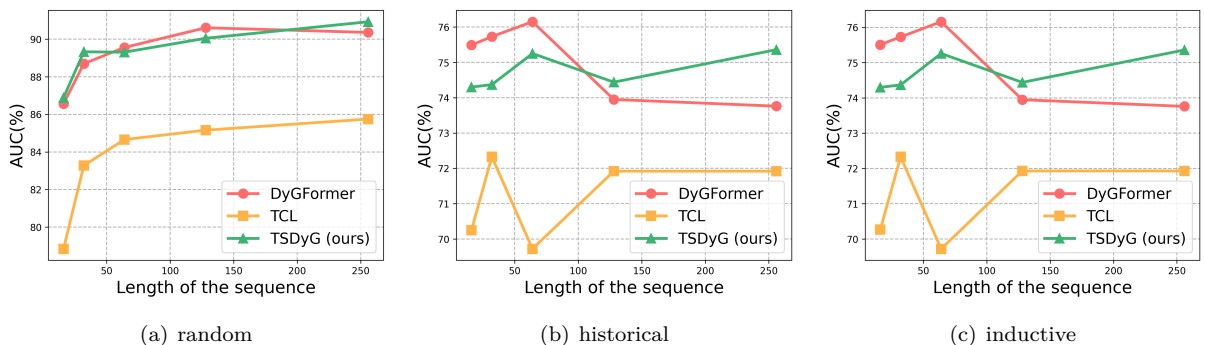

(a) random         (b) historical         (c) inductive

Figure 9: Performance of our method with different sequence lengths ranging from 32 to 256 under random, historical, and inductive sampling strategies. The evaluations are performed on Wikipedia.

