# OpenReview forum: "Revisiting Dynamic Graphs from the Perspective of Time Series"
_TMLR — Under review for TMLR_

### Review · Reviewer_5uqJ · 2026-06-15

**Summary Of Contributions:**

This paper provides a unique approach for dynamic graph representation, which includes the pair-level time-series, timestamp series, edge feature series and co-occurence frequency. This representation can be viewed as key details extracted from the original dynamic graphs. The most significant contribution is the model architecture in the proposed method fully leverages this dynamic graph representation and achieves great performance in extensive experiments.

Strengths:
1. The proposed model architecture is clearly illustrated, and the design is effective and scalable.
2. This paper includes extensive experiments to evaluate the performance of the TSDyG method, with 14 different benchmarks and 10 baseline methods. The details of the experiments in clear enough for reproduction.
3. The computational cost analysis is compelling. It shows the scalability of the method to long sequence with efficiency.

Weakness:
1. The Eq.1 is incomplete. It fails to make definition of $f_{u,v}$ for all $(u,v)$ pairs. For example, if both u and v have no interaction history with any node, what would $f_{u,v}$ be?
2. The content of Figure 2 does not seem to correspond well with the Preliminary section, and Figure 2(c) is difficult to understand, making it is difficult to intuitively find out the key differences between the three dynamic graphs.
3. It is not necessary to attribute the effectiveness of the method to its ability to capture periodic interactions between nodes, unless further direct evidence is provided.

**Audience:**

Yes

**Audience Explanation:**

This paper provides a new effective method to analyze dynamic graphs, which is one of the top research interests of some TMLR audience.

**Broader Impact Concerns:**

I do not have any ethical concerns.

**Claims And Evidence:**

Yes

**Claims Explanation:**

The experiments in this paper is clear and extensive. It would be better to provide some evaluation on the ability to capture periodic interactions.

**Requested Changes:**

I believe it is necessary to redraw Figure 2 to make the proposed TSG more clear. Including a simulation study with different level of recurrent or periodic interaction and showing the effectiveness of the method in capturing them would be extremely helpful.

---

> ### Author Response · Authors · 2026-07-14
> **Response**
>
> **Q1. The Eq.1 is incomplete.**
>
> Thank you for pointing this out. We have revised Eq. (1) to explicitly account for timestamps at which neither uuu nor vvv participates in an interaction. The updated Eq. (1) is given as follows:
> $
> f_{u \rightarrow v}(t)=
> \begin{cases}
> 1, & \text{if } (u,v,t)\in\mathcal{G}
> \text{ or } (v,u,t)\in\mathcal{G},\\
> 0, & \text{otherwise}.
> \end{cases}
> $
> We will update it in our revised manuscript.
>
> **Q2. The content of Figure 2 does not seem to correspond well with the Preliminary section, and Figure 2(c) is difficult to understand,**
>
> Thank you for this helpful suggestion. We agree that the original Figure 2 did not align clearly enough with the description in the Preliminary section and that Figure 2(c) was not sufficiently intuitive in illustrating the proposed time series-based graph representation. We will redraw Figure 2 in the revised manuscript .
>
> **Q3. Including a simulation study with different level of recurrent or periodic interaction and showing the effectiveness of the method in capturing them would be extremely helpful.**
>
> Thank you for this valuable suggestion. To directly examine whether our model can capture recurring and periodic interaction patterns, we conduct a controlled simulation study on synthetic dynamic graphs. We generate a synthetic dataset with 100 nodes and 20 target node pairs, where each pair follows a periodic interaction pattern with a randomly assigned phase offset. We also add non-periodic background interactions among other node pairs to avoid making pair identities trivially predictive.
>
> We control the strength of temporal regularity using a parameter $\alpha \in \{0, 0.5, 1.0\}$. When $\alpha=0$, interactions are temporally random; when $\alpha=0.5$, periodic patterns are mixed with random variation; and when $\alpha=1.0$, interactions exhibit strong periodicity. The average interaction density is kept approximately constant across the three settings so that the primary difference is the level of temporal regularity.
>
> For each periodicity setting, we generate 30 periods of interactions and apply the same chronological 70%/15%/15% training, validation, and test split used in the main experiments. Each experiment is repeated using three random seeds. We compare TSDyG with DyGFormer under random, historical, and inductive negative-sampling settings.
>
> As shown in Table 1, TSDyG consistently outperforms DyGFormer across all periodicity levels and negative-sampling settings. More importantly, the performance gap generally increases as the periodicity strength α becomes larger. Under random negative sampling, the improvement increases from 0.45 AP points at $\alpha=0$ to 1.06 points at $\alpha=1.0$. Similarly, the improvement grows from 2.08 to 2.90 points under historical sampling and from 1.12 to 2.55 points under inductive sampling. These results provide controlled supporting evidence that the proposed pair-conditioned time-series formulation can effectively exploit recurring and periodic interaction patterns. We will include the simulation setup, results, and corresponding discussion in the revised manuscript.
>
> **Table 1.** Average precision performance of DyGFormer and TSDyG on a synthetic dataset with varying levels of temporal regularity under random, historical, and inductive negative-sampling settings.
>
> | NSS | Model | $\alpha = 0$ | $\alpha = 0.5$ | $\alpha = 1.0$ |
> |---|---|---|---|---|
> | rnd | DyGFormer | 93.82±0.04 | 94.51±0.02 | 96.70±0.01 |
> | rnd | TSDyG (Ours) | **94.27±0.03** | **95.84±0.03** | **97.76±0.01** |
> | his | DyGFormer | 83.26±0.45 | 84.90±0.36 | 85.96±0.27 |
> | his | TSDyG (Ours) | **85.34±0.37** | **87.63±0.41** | **88.86±0.25** |
> | ind | DyGFormer | 81.46±0.50 | 83.08±0.33 | 84.48±0.23 |
> | ind | TSDyG (Ours) | **82.58±0.53** | **84.87±0.39** | **87.03±0.18** |

---

> > ### Comment · Reviewer_5uqJ · 2026-07-22
> >
> > Thanks for the point-to-point response. My concerns are addressed.

---

### Review · Reviewer_pGku · 2026-06-27

**Summary Of Contributions:**

**Summary:**

This paper revisits the dynamic graph learning problem from a time series perspective. The main idea is to represent the interaction history of each source-destination node pair as a binary time series, so that recurring and long-range temporal patterns can be modelled explicitly. Building on this formulation, the authors propose Time Series-based Dynamic Graph (TSDyG), which combines the binary-series-represented dynamic graph with timestamp, edge, and co-occurrence embeddings, and feeds them into a cross-attention module that uses a single learnable query token, focusing on the future link prediction task. Experiments cover multiple benchmarks and settings, reporting competitive performance of TSDyG compared to strong baselines.

**Contributions:**

1. A novel time series formulation for dynamic graphs, representing the interaction history of each source-destination node pair as a binary series to expose temporal regularities.
2. The proposed TSDyG model is built on this formulation, concatenating the binary series with various structural/temporal embeddings and feeding them into a cross-attention module for efficient long-sequence modelling.
3. Extensive experiments across multiple benchmarks show that TSDyG achieves competitive performance against several strong baselines on the link prediction task.

**Strengths:**

- [S1] The time series reformulation is intuitive and offers a fresh perspective on dynamic graph learning. Casting each source-destination pair's interaction history as a binary series is a simple way to expose recurring patterns.
- [S2] The empirical evaluation is broad, covering two benchmark collections across diverse domains, together with ablations and a computational-cost analysis. All results are reported with standard deviations over five runs, which supports the reliability of the comparisons.
- [S3] The paper is well written, well structured, and easy to follow. The figures and tables effectively illustrate key points and results.

**Weaknesses:**

- [W1] The "superior performance" claim of TSDyG is slightly overstated. Under the random setting, the gains over the strongest baseline are sometimes marginal. Besides, the AP average rank is tied with DyG-Mamba at 1.50 in Table 1, and several bold-marked "best" results overlap with the second-best under standard deviation. The clear improvements mainly appear in the historical and inductive settings.
- [W2] The ablation study is a little narrow. It only removes the input features (binary series, timestamps, edge, and co-occurrence) but never isolates the cross-attention module, which is the core design of the model. Although CATS is compared as a standalone baseline in Table 3, there is no controlled comparison that replaces the single-query cross-attention with, e.g., self-attention or a simple pooling, so its individual contribution remains unclear.

**Additional Comments:**

none

**Audience:**

Yes

**Audience Explanation:**

Yes. There are at least some individuals in TMLR's audience who work on dynamic graphs and temporal link prediction and would be interested in the findings of this paper. The time series reformulation for dynamic graphs is a simple, transferable idea, and the demonstration that standard time series models can be adapted to dynamic link prediction would also interest the time series community.

**Broader Impact Concerns:**

No broader impact concerns. The work uses public benchmarks for the link prediction task and introduces no new data or capabilities that raise ethical concerns.

**Claims And Evidence:**

Yes

**Claims Explanation:**

Yes. The paper's main claims, a time-series reformulation of dynamic graphs and competitive link-prediction performance, are supported within the authors' stated scope, with accurate, convincing, and clear evidence.

**Requested Changes:**

- **[Strengthening]** The "superior performance" wording is slightly overstated under the random setting, where some gains over the strongest baseline fall within one standard deviation. Scoping these claims to the historical and inductive settings, or reporting a paired t-test with p-values on a representative subset to confirm the gains are statistically significant, would make the comparison more convincing.
- **[Strengthening]** The ablation study could isolate the cross-attention module, which is the core design of the model. A controlled comparison replacing the cross-attention with other methods/modules would clarify its individual contribution.

---

> ### Author Response · Authors · 2026-07-14
> **Response**
>
> **Q1. The "superior performance" wording is slightly overstated under the random setting, where some gains over the strongest baseline fall within one standard deviation. Scoping these claims to the historical and inductive settings,**
>
> Thank you for this valuable suggestion. We agree that the term “superior performance” is too broad, particularly under random negative sampling, where the improvements over the strongest baselines are often modest and may fall within one standard deviation. We will revise the manuscript to qualify these claims more carefully and to accurately reflect the empirical results.
>
> **Q2. The ablation study could isolate the cross-attention module, which is the core design of the model. A controlled comparison replacing the cross-attention with other methods/modules would clarify its individual contribution.**
>
> Thank you for this valuable suggestion. To investigate the contribution of the cross-attention module, we replace it with full self-attention and simple pooling while keeping all other components unchanged. We evaluate the three variants on the Wikipedia and Enron datasets under random, historical, and inductive negative-sampling settings. The corresponding average precision results are reported in Table 1 below.
>
> The results show that the proposed cross-attention module achieves better performance than full self-attention and simple pooling on most evaluation settings. In addition, compared with full self-attention, cross-attention provides lower computational complexity while maintaining, and in most cases improving, predictive performance. These findings demonstrate that the proposed module offers a favorable balance between effectiveness and efficiency. We will include this additional ablation study and the corresponding discussion in the revised manuscript.
>
> **Table 1.** Average precision performance of TSDyG using cross-attention, full self-attention, and simple pooling on multiple benchmark datasets under random, historical, and inductive negative-sampling settings.
>
> | NSS | Method | Wikipedia | Enron |
> |---|---|---:|---:|
> | rnd | TSDyG (pooling) | 99.81±0.01 | 91.98±0.03 |
> | rnd | TSDyG (self-attention) | **99.20±0.01** | **92.84±0.21** |
> | rnd | TSDyG (cross-attention) | **99.20±0.01** | 92.56±0.18 |
> | his | TSDyG (pooling) | 71.79±0.35 | 77.25±0.58 |
> | his | TSDyG (self-attention) | 83.13±0.40 | 79.57±0.70 |
> | his | TSDyG (cross-attention) | **87.52±0.36** | **80.26±0.66** |
> | ind | TSDyG (pooling) | 77.07±0.30 | 78.93±0.10 |
> | ind | TSDyG (self-attention) | 84.14±0.25 | **81.13±0.25** |
> | ind | TSDyG (cross-attention) | **87.23±0.27** | 80.51±0.12 |

---

> > ### Comment · Reviewer_pGku · 2026-07-21
> >
> > Thank you very much for your response, which has addressed most of my concerns.

---

### Review · Reviewer_9y72 · 2026-06-29

**Summary Of Contributions:**

This paper studies dynamic graph learning from a time-series perspective. The authors argue that DTDG methods lose fine-grained temporal information due to snapshot discretization, while CTDG methods may be less effective at capturing long-range and recurring temporal patterns. To address this, the paper proposes TSDyG, which converts historical interactions between node pairs into binary time-series representations and models them with an embedding generation module and a cross-attention module.

**Audience:**

Yes

**Audience Explanation:**

This paper addresses a research problem of significant value and proposes a method with certain innovative elements to solve this problem.

**Claims And Evidence:**

Yes

**Claims Explanation:**

The experimental results sufficiently support the claims made in the paper.

**Requested Changes:**

1. Insufficient research motivation innovation: The research motivation of this paper lacks innovation, as similar research motivations have been explicitly proposed in works such as FreeDyG. It is recommended that the authors re-examine and clearly define the unique contributions and innovative value of this paper compared to existing works.

[1]Tian, Yuxing, Yiyan Qi, and Fan Guo. "Freedyg: Frequency enhanced continuous-time dynamic graph model for link prediction." The twelfth international conference on learning representations. 2024.

2. The author mentions that the problem that “selecting an appropriate snapshot interval is non-trivial” exists in DTDG methods, where sampling at different time intervals may reveal distinctly different temporal patterns and inherent regularities, and how to set the values is crucial. However, the same problem exists in this paper.
3. For periodic patterns, if only discrete time points are sampled, how can this work ensure that the sampling captures the periodic regularities? Furthermore, for edges that occur infrequently and are relatively sparse, could the current sampling strategy miss such edges?
4. Considering the significant advantages of frequency-domain methods in capturing periodic patterns in time series, it is recommended to supplement relevant comparisons to comprehensively verify the effectiveness and competitiveness of the proposed method. (Such as works combining graph and frequency-domain methods like FreeDyG and UniDyG.)

[1]Tian, Yuxing, Yiyan Qi, and Fan Guo. "Freedyg: Frequency enhanced continuous-time dynamic graph model for link prediction." The twelfth international conference on learning representations. 2024.

[2]Xu, Yuanyuan, et al. "Unidyg: a unified and effective representation learning approach for large dynamic graphs." IEEE Transactions on Knowledge and Data Engineering (2025)

---

> ### Author Response · Authors · 2026-07-14
> **Response**
>
> **Q1. It is recommended that the authors re-examine and clearly define the unique contributions and innovative value of this paper compared to existing works.**
>
> Thank you for raising this important concern. We agree that FreeDyG also recognizes recurring and periodic interaction patterns in dynamic graphs and is motivated by capturing the temporal “shift” phenomenon in evolving interaction patterns.  Although both FreeDyG and our method aim to exploit temporal regularities, they differ fundamentally in their problem formulation, input representation, and modeling mechanism.
> First, FreeDyG remains within the conventional continuous-time dynamic graph framework, where historical interactions are represented as chronologically ordered event sequences. In contrast, our work introduces an explicit pair-conditioned time-series formulation of dynamic graphs. This formulation characterizes the evolution of a specific node pair by distinguishing their mutual interactions from interactions involving alternative nodes. As a result, it provides a more direct representation of how the relationship between a target node pair develops over time relative to competing interactions.
>
> Second, the two methods capture temporal regularities through fundamentally different mechanisms. FreeDyG models evolving and shifting interaction patterns primarily through frequency-domain enhancement. In contrast, our method does not rely on Fourier transformation or frequency filtering. Instead, it directly models the ordered, pair-specific interaction sequence in the time domain by distinguishing interactions within the target node pair from interactions involving alternative nodes. This design makes the temporal regularities and evolving interaction patterns between a target pair of nodes more explicit.
>
> Finally, a distinguishing contribution of our work is that it offers a fresh perspective on dynamic graph learning. Beyond the proposed TSDyG model, our time-series formulation broadens the methodological scope of dynamic graph research and establishes a meaningful bridge between dynamic graph learning and time-series modeling. We will revise the motivation and contribution statements of our manuscript accordingly.
>
> **Q2 The author mentions that the problem that “selecting an appropriate snapshot interval is non-trivial” exists in DTDG methods,  However, the same problem exists in this paper.**
>
> Thank you for pointing this out. We would like to emphasize that our method does not partition the interaction stream into fixed-width temporal snapshots. Instead, for a target pair (u,v), it constructs a pair-conditioned sequence from all actual historical events involving the source node u and all of its neighbors, including the target node v. The model also preserves the fine-grained temporal information associated with each historical event through explicit temporal encoding. Therefore, our method does not require selecting a snapshot interval or alters the temporal resolution of the original dynamic graph. We will revise the manuscript to make it more clear.
>
> **Q3. For periodic patterns, if only discrete time points are sampled, how can this work ensure that the sampling captures the periodic regularities? Furthermore, for edges that occur infrequently and are relatively sparse, could the current sampling strategy miss such edges?**
>
> Thank you for raising this question. We would like to clarify that our method does not sample the interaction history at uniformly spaced discrete time points. Instead, it constructs the sequence from all actual events of historical events involving the target nodes. Therefore, no interaction is discarded merely because it falls between predefined sampling points, and the irregular time intervals between events are explicitly preserved through the time encoder. This event-driven construction allows the model to retain fine-grained temporal information and identify recurring patterns from both the ordering of interactions and their associated time intervals.

---

> > ### Author Response · Authors · 2026-07-14
> > **Response 2**
> >
> > **Q4. it is recommended to supplement relevant comparisons to comprehensively verify the effectiveness and competitiveness of the proposed method.**
> >
> > Thank you for the suggestion. Table 1 below reports the average precision of TSDyG and FreeDyG across multiple benchmark datasets under random, historical, and inductive negative-sampling strategies. The results show that our method achieves competitive performance compared with FreeDyG across these evaluation settings. In the revised manuscript, we will provide a more explicit comparison and include additional results.
> >
> > **Table 1.** Average precision performance of TSDyG and FreeDyG on multiple benchmark datasets under random, historical, and inductive negative-sampling settings.
> >
> > | NSS | Method | Wikipedia | LastFM | Enron |
> > |---|---|---:|---:|---:|
> > | rnd | FreeDyG | **99.26±0.01** | 92.15±0.16 | 92.51±0.05 |
> > | rnd | TSDyG (Ours) | 99.20±0.01 | **93.64±0.11** | **92.56±0.18** |
> > | his | FreeDyG | **91.59±0.57** | 79.71±0.51 | 78.87±0.82 |
> > | his | TSDyG (Ours) | 87.52±0.36 | **84.04±0.35** | **80.26±0.66** |
> > | ind | FreeDyG | **90.05±0.79** | **72.19±0.24** | 77.81±0.65 |
> > | ind | TSDyG (Ours) | 87.23±0.27 | 71.69±1.06 | **80.51±0.12** |